Methods

# Tracking connectivity maps in human stem cell–derived neuronal networks by holographic optogenetics

Felix Schmieder[1],* , Rouhollah Habibey[2],* , Johannes Striebel[2] , Lars Büttner[1] , Jürgen Czarske[1,3,4,5] , Volker Busskamp[2]

**Neuronal networks derived from human induced pluripotent stem cells have been exploited widely for modeling neuronal circuits, neurological diseases, and drug screening. As these networks require extended culturing periods to functionally mature in vitro, most studies are based on immature networks. To obtain insights on long-term functional features, we improved a glia–neuron co-culture protocol within multi-electrode arrays, facilitating continuous assessment of electrical features in weekly intervals. By full-field optogenetic stimulation, we detected an earlier onset of neuronal firing and burst activity compared with spontaneous activity. Full-field stimulation enhanced the number of active neurons and their firing rates. Compared with full-field stimulation, which evoked synchronized activity across all neurons, holographic stimulation of individual neurons resulted in local activity. Single-cell holographic stimulation facilitated to trace propagating evoked activities of 400 individually stimulated neurons per multi-electrode array. Thereby, we revealed precise functional neuronal connectivity motifs. Holographic stimulation data over time showed increasing connection numbers and strength with culture age. This holographic stimulation setup has the potential to establish a profound functional testbed for in-depth analysis of human-induced pluripotent stem cell-derived neuronal networks.**

## Introduction

Complex neuronal circuits of the human brain are composed of diverse neuronal cell types and their elaborate connections (Bassett & Sporns, 2017; Kriegeskorte & Douglas, 2018). Determining how a single neuron contributes to circuit function and behavior is a key step towards unraveling functional connectivity maps within neuronal circuits (Lim et al, 2013; Stam, 2014; Dal Maschio et al, 2017).

The emerging field of human pluripotent stem cell technology enables bottom-up neuroscientific approaches by generating neurons and neuronal networks in vitro. Especially for biomedical studies, this is complementary to animal model systems and has the potential to reduce the translational gap from bench-to-bedside (Bassett & Gazzaniga, 2011; Ardhanareeswaran et al, 2017; Latifi et al, 2020). Human embryonic stem cells (hESCs) (Ilic & Ogilvie, 2017) and human-induced pluripotent stem cells (hiPSCs) (Hockemeyer & Jaenisch, 2016; Shi et al, 2017) offer an almost unlimited cell source for neuronal cell and circuit engineering. Many protocols to drive stem cells into neurons require multiple steps and long time periods using soluble factors and specific culturing techniques (Sauter et al, 2019). There exist also single-step differentiation protocols by inducible neurogenic transcription factor expression that result in rapid neurogenesis (Pang et al, 2011; Zhang et al, 2013; Busskamp et al, 2014). For example, the activation of neurogenin-1 and neurogenin-2 by the TetOn promoter system in hiPSCs and hESCs, independent of the underlying stem cell line, triggers neurogenesis, leading to postmitotic neurons in just 4 days post induction (dpi). The inducible neurogenin expression cassette was stably integrated into hiPSCs, the so-called iNGN cells (Busskamp et al, 2014).

To date, the electrophysiological properties of most human stem-cell-derived neurons at the circuit level are insufficiently characterized. Most in vitro neuronal cultures mature slowly (Odawara et al, 2014; Amin et al, 2016; Chevée & Brown, 2018). Therefore, neuronal activity is frequently induced by current injection at a few time points to study some functional aspects (Amin et al, 2016). Continuous probing of the developing human neuronal circuit activity with patch-clamp recordings is challenging as one can only record once from the same cell. Also, sterility in the recording chamber cannot be guaranteed, requiring the disposal of the cultures after one experiment (Sakmann & Neher, 1984). Multi-electrode arrays (MEAs), on the other hand, facilitate recordings from the same hiPSC-derived neuronal networks or primary neuronal cultures over time through their non-invasive and sterile

[1]Laboratory of Measurement and Sensor System Technique, Faculty of Electrical and Computer Engineering, TU Dresden, Dresden, Germany   [2]Department of Ophthalmology, Universitäts-Augenklinik Bonn, University of Bonn, Bonn, Germany   [3]Competence Center for Biomedical Computational Laser Systems (BIOLAS), TU Dresden, Dresden, Germany   [4]Cluster of Excellence Physics of Life, TU Dresden, Dresden, Germany   [5]Institute of Applied Physics, School of Science, TU Dresden, Dresden, Germany

Correspondence: juergen.czarske@tu-dresden.de; volker.busskamp@ukbonn.de
*Felix Schmieder and Rouhollah Habibey contributed equally to this work.

working mode (Massobrio et al, 2015; Saalfrank et al, 2015; Latifi et al, 2016; Habibey et al, 2017; Kizner et al, 2019). For instance, long-term activity profiles of the commercially available hiPSC-derived "iCell" neurons were studied using standard MEAs (Odawara et al, 2013; Lu et al, 2019) and high-density CMOS-based MEAs (Amin et al, 2016). These studies have revealed increased neuronal firing rates, that is, action potential (AP) frequencies, over time (Lu et al, 2019), whereas the earliest detection of synchronized burst activity with higher latencies was at day 70 of culture (Amin et al, 2016; Odawara et al, 2016). However, long-term network functional connectivity features were not revealed as electrical stimulation of individual neurons through MEA electrodes is challenging. Stimulation artifacts impede the extraction of data during and after the applied electrical pulses through the recording electrodes (Wagenaar et al, 2004; Mena et al, 2017). Low density neuronal cultures on high-density MEAs enabled to electrically trigger individual neuronal activity (Bakkum et al, 2013); however, electrical stimulation of single neurons in densely connected networks is often challenging. High-density MEA substrates are not transparent, which limits live imaging of morphological details at single neurons resolution. In addition, neuronal activity cannot be inhibited by MEA electrodes (Wagenaar et al, 2004). Here, optogenetics is advantageous, providing precise spatial and temporal resolution to excite or inhibit neuronal activity (Lignani et al, 2013). Channelrhodopsin-2 (ChR2) is among the most used optogenetic actuators for inducing neuronal activity (Deisseroth, 2011; Guru et al, 2015; Habibey et al, 2020). Focused or patterned light stimuli enable control of the activity of specific neurons within a defined neuronal network (Mohanty & Lakshminarayananan, 2015). Optogenetics has already been widely exploited for mapping microcircuits by targeting subsets of neurons in brain slices in vitro (Weiler et al, 2008). Optogenetic-based probing of in vivo inter-regional and global connections within rodent brain areas is also commonly used (Ayling et al, 2009; Lim et al, 2012). Large areas of rat primary cortical networks have been functionally studied, in combination with optogenetic manipulation, to extract functional connectivity maps (Saber et al, 2018). However, optogenetic-based mapping of functional connectivity has not been explored in human stem cell–derived neuronal networks in vitro. Most functional studies on hiPSC-derived neurons provide temporary information on network activity levels, whereas in-depth functional analysis of the in vitro neuronal network development remains elusive (Lam et al, 2017; Hyvärinen et al, 2019).

Advanced optogenetic measurements use two-photon stimulation to penetrate deep into the neuronal tissues both ex vivo (Yang et al, 2018) and in vivo (Dalgleish et al, 2020). Unfortunately, two-photon stimulation setups are unaffordable for many labs. Single-photon stimulation methods, for example, galvanometric actuator or acousto-optic deflector scanning techniques, are not able to stimulate several neurons simultaneously (Wang et al, 2007, 2011). Other approaches like arrays of microscopic light-emitting diodes (LEDs) (Morton et al, 2019; Meloni et al, 2020) and needle-like optrodes (Goncalves et al, 2018) suffer from poor spatial resolution or heat generation (Goncalves et al, 2018). Pixelated light modulation devices with millions of individually addressable actuators like digital mirror devices (DMDs) or spatial light modulators (SLMs) grant more flexible modulation of light amplitude or phase to create stimulation patterns on the sample. DMDs can reach switching rates of several kHz but suffer from poor light or diffraction efficiency (Ronzitti et al, 2017; Shemesh et al, 2017). Stimulation with computer-generated phase holograms is more flexible even without additional devices (Papagiakoumou, 2013): It enables higher stimulation energies than DMD and organic light-emitting diodes, three-dimensional scanning (Ronzitti et al, 2017), versatile fiber-optic approaches (Büttner et al, 2020; Tehrani et al, 2021), and compensation of sample-induced aberrations (Tehrani et al, 2021). Holographic optogenetic stimulation with single-cell spatial resolution and sufficient temporal precision, less than a millisecond, has been applied to brain slices and intact brain tissue (Ronzitti et al, 2017; Shemesh et al, 2017; Gill et al, 2020). Whereas holographic stimulation of in vivo networks offers great potential for mapping of functional circuits, this approach has not yet been applied to in vitro developing neuronal networks.

Here, we implemented a holographic stimulation device based on a ferroelectric phase-modulating SLM with switching rates of up to 400 Hz for MEA recordings (Schmieder et al, 2018). Automatic calibration facilitated precise targeting and activation of individual hiPSC-derived neurons with high spatial resolution (8 $\mu$m) and sufficient temporal resolution (2.5 ms) at high stimulation energies (0.15 W/mm$^2$). Continuous weekly recordings over 3 mo generated precise connectivity data from developing human neuronal networks. Our optimized human neuronal culturing protocol mediated long-term survival of the neuronal cultures for stable holographic stimulation experiments over hours. The holographic stimulation of single optogenetically tagged neurons, on average 400 per MEA culture, evoked robust neuronal activity. This cellular one-by-one stimulation, in contrast to commonly used full-field light stimulation, facilitated the derivation of dynamic individual connectivity motifs, including single-neuron connectivity patterns, total neuronal connection numbers, and synaptic strengths. Extracting these functional features from individual neurons within random neuronal networks is important to exploit these in vitro model systems more widely for both basic and biomedical neuroscience research.

## Results

### Stable co-culture platform for long-term optogenetic experiments on human stem cell–derived neurons

To continuously record neuronal activity from iNGN neurons over months, we have established an advanced culturing protocol on MEAs (Fig 1A–C, see the Materials and Methods section). Upon neuronal induction, iNGN cells were transduced with lentiviral particles delivering ChR2-EYFP driven by the ubiquitous elongation factor 1α (ef1α) promoter element. The fused EYFP tag resulted in membrane-bound labeling of targeted neurons. Neuronal morphologies were assessed by fluorescent live cell imaging within the MEA setup. The antimetabolic agent cytosine arabinoside (Ara-C) was applied at 4 dpi to remove potential undifferentiated hiPSC cells that would otherwise grow over the postmitotic neurons over time. At 5 dpi, the differentiated iNGN cells were enzymatically detached from the culturing plate and reseeded onto the MEAs

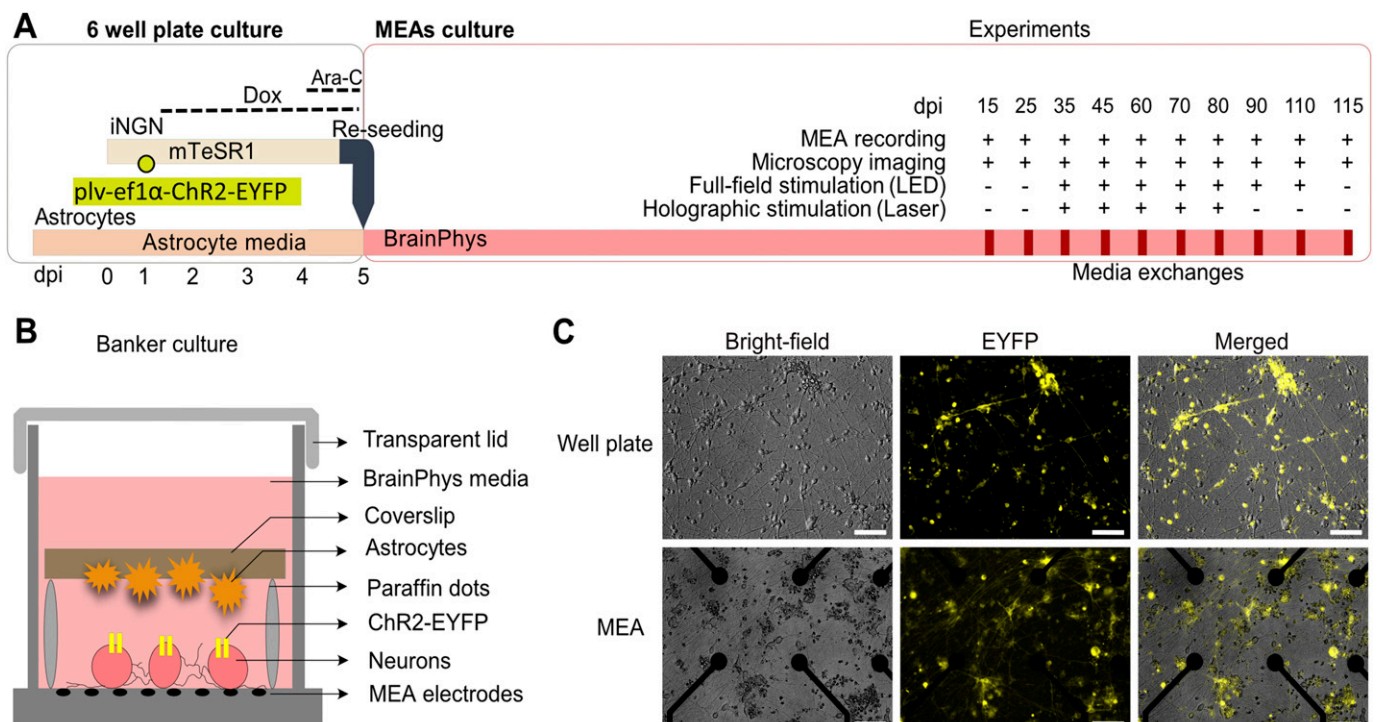

**Figure 1. iNGN–astrocyte co-culture system on transparent multi-electrode arrays (MEAs).**
**(A)** Experimental scheme of iNGN–astrocyte co-culture preparation (left) and experimental procedure (right) over time. Functional recordings, microscopy imaging, full-field, and holographic stimulations were studied on different days post induction (dpi). **(B)** Schematic of the so-called Banker culture, a neuron–glia co-culture system within the MEA device. **(C)** Representative microscopy images of iNGN cells. Upper panel, iNGN cells grown on Matrigel-coated well plates at 3 dpi. Lower panel, PDL-laminin–coated MEA chambers at 15 dpi. ChR2-EYFP–labeled cells are visualized by live cell microscopy. Scale bars, 100 $\mu$m. mTeSR1: standardized human induced pluripotent stem cell medium. BrainPhys: serum-free neurophysiological basal medium for improved neuronal function. plv-ef1α-ChR2-EYFP: lentiviral particles delivering ChR2-EYFP under the ef1α promotor to iNGN cells. Dox: doxycycline. Ara-C: cytosine arabinoside to remove undifferentiated cells.

(Fig 1A). Astrocytes were added upside-down on coverslips, separated by paraffin spacers from the neuron layer (Fig 1B).

These so-called Banker neuron–glia co-cultures (Kaech & Banker, 2006), in combination with neuronal media and sterile membranous lids, enabled long-term cultivation and continuous MEA recordings. For each recording session, an MEA culture was placed into the MEA setup and 3 min of baseline recording, 5 min of full-field and 1 h holographic light stimulation was performed (Fig 1A). All full-field and holographic optogenetic experiments were performed with MEAs that were sealed with transparent caps to prevent media evaporation and contaminations, while enabling fluorescent imaging and optical stimulations (Figs 1B and S1). Altogether, our optimized protocol facilitated continuous functional recordings covering the maturation and neuronal circuit development periods of hiPSC-derived neurons.

### Neuronal activity evoked by full-field optogenetic stimulation

Spontaneous neuronal activity of hiPSC-derived neurons is considered a sign of neuronal maturation, which can take weeks to months in vitro. In some electrodes, we detected signs of spontaneous iNGN activity at 25 dpi that increased over time and reached an average frequency peak of 0.89 ± 0.14 Hz at 60 dpi (Figs 2 and S2C). At later time points, the AP frequency decreased again to 0.2 ± 0.04 Hz at 70 dpi and 0.51 ± 0.10 Hz at 80 dpi (Fig S2C). Electrode

data were sorted to identify individual neurons (Fig 2E). Firing rates of individual neurons also showed similar peaks at 60 dpi followed by a decline (0.49 ± 0.13 and 0.17 ± 0.04, respectively; Fig 2F). Developing neuronal networks typically have a peak in activity followed by a decline (Chiappalone et al, 2006), suggesting that human stem-cell-derived neuronal networks replicate this developmental feature in vitro.

Next, we compared spontaneous and optogenetically evoked activities (N = 4 MEAs and n = 95 active electrodes, from 15 to 80 dpi). Full-field light stimulation (470 nm, 50 ms pulses, 0.5 Hz, 0.46 mW/mm$^2$; Fig S10) induced APs starting at 15 dpi when unstimulated neurons were inactive ($P < 0.05$ versus baseline, Fig S2C). The signal profile and activity raster plot showed coupled responses to blue light pulses (Figs 2A and S2A and B). Between 35 and 80 dpi, full-field stimulation significantly boosted network activity compared with spontaneous activity ($P < 0.01$ versus baseline Fig S2C). The average AP frequency of light-ON and light-OFF intervals in electrodes reached 2.69 ± 0.35 Hz at 60 dpi (Fig S2C). As revealed by spike sorting, individual electrodes captured the activity of several neurons (Figs 2E and S3A and B). The AP waveforms were similar between spontaneous and light-induced activity (Figs 2E and S3). Probing the activity in individual neurons also showed coupled responses to the applied light pulses (Fig 2B and D). When full-field optogenetic stimulation was applied to the entire array area, more neurons were active than the non-stimulated condition at baseline

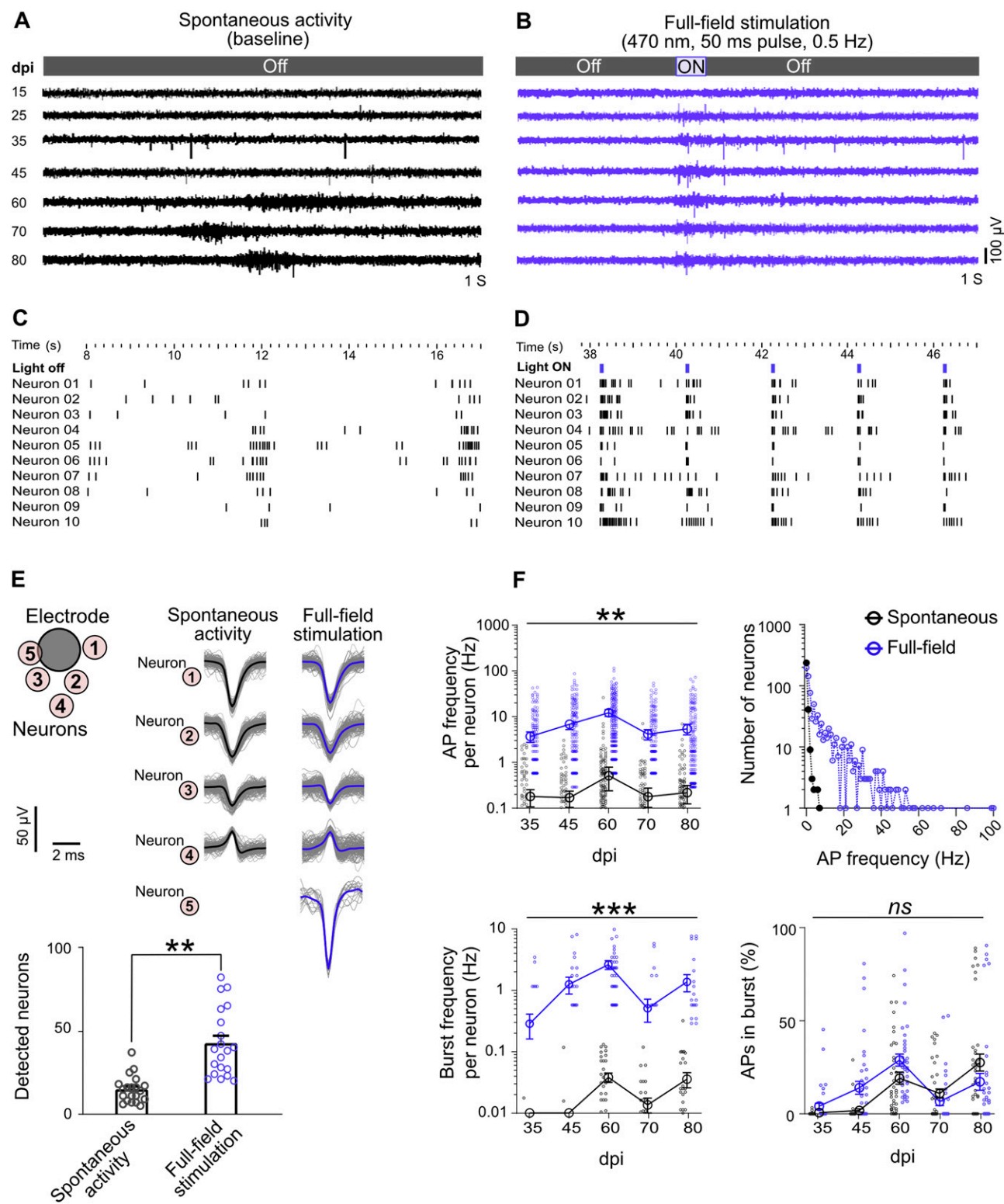

**Figure 2. Full-field optogenetic stimulation elevated network activity features (blue) compared with spontaneous activity (black).**
**(A)** Spontaneous activity from a representative electrode over time, from 15 to 80 days post induction (dpi). **(B)** Full-field stimulation: optically evoked activity in the same electrode on different days. **(C)** Raster plot of detected action potentials in 10 neurons on 60 dpi (also see Fig S2). **(D)** Raster plot of neuronal responses to 0.5 Hz light pulses of full-field stimulation (60 dpi). **(E)** The AP waveforms of several neurons were extracted by spike sorting in one electrode. Neuron 5 was only active during optogenetic stimulation. Bars show the average number of detected neurons per culture (n = 4 multi-electrode arrays) in stimulated and non-stimulated conditions (**$P < 0.01$, Wilcoxon test). Data pooled from 35 to 80 dpi. **(F)** Spontaneous and optogenetically evoked activity features in neuronal units. AP frequency per neuron

(42.5 ± 4.5 versus 14.85 ± 1.79 neurons; Fig 2E). The firing rates of the sorted neurons at all days were elevated by full-field stimulation (Fig 2F). Whereas the spontaneous firing rates of all neurons were below 10 Hz, full-field optogenetic stimulation shifted neuronal firing rates to higher frequencies up to 100 Hz (Fig 2F).

The appearance of AP bursts is an important characteristic of neuronal network maturation (Suresh et al, 2016). Spontaneous burst activity was only captured by few electrodes starting at 35 dpi and reached its peak at 60 dpi. Light-evoked burst activity was already detected at 25 dpi (P < 0.05 versus baseline; Fig S2D), increased over time, and reached its peak at 60 dpi (P < 0.001 versus 15 and 25 dpi, and P < 0.01 versus baseline; Fig S2D). The average burst duration remained stable over time and was not different between spontaneous and optogenetic-evoked activity (Fig S2E). The percentage of APs appearing in bursts rather individually increased with culture age (Fig S2F). Burst activity was also studied in individual neurons (Fig 2F), which showed elevated burst frequencies during optogenetic stimulation (P < 0.001 versus spontaneous bursts; Fig 2F). The percentage of neuronal APs occurring in burst activity did not change significantly with optogenetic stimulation (Fig 2F).

To distinguish between directly stimulated neurons and responses resulting from synaptically connected functional circuits, we applied pharmaceutical inhibition. We added the glutamate receptor blocker NBQX (2,3-dioxo-6-nitro-7-sulfamoyl-benzo[f] quinoxaline) and APV ((2R)-amino-5-phosphonovaleric acid) to the cultures, which resulted in a decrease in APs and burst frequencies recorded by electrodes (0.80 ± 0.09 Hz and 0.04 ± 0.01 versus 0.003 ± 0.001 Hz and 0.00 ± 0.00, respectively, P < 0.001 versus baseline; Fig S4A and B). Isolated neuronal units also became silent in the presence of NBQX-APV (Fig S5B and C). Full-field optogenetic stimulation in presence of NBQX-APV increased AP frequency in few electrodes (0.04 ± 0.02 Hz, Fig S4A and B) and in limited number of neurons (Fig S5B and C). Even higher intensities of full-field stimulation were not able to return synchronized burst activity in the presence of NBQX+APV (P < 0.001 versus baseline in untreated conditions; Fig S4A). After washout, AP frequencies increased under both baseline and light-stimulated conditions in electrodes (3.42 ± 73 and 3.73 ± 0.8, respectively, P < 0.05 versus before treatment values; Fig S4A). Burst activity also reset after washout under light-stimulated and baseline conditions (Figs S4B and S5C). Peri-stimulus time histograms (PSTH) were plotted for the electrode data and individual neurons that responded to the full-field stimulation under NBQX-APV treatment (Figs S4C and S5D). PSTH in baseline are composed of an early peak during the stimulus followed by an extended response during light-off period (Fig S4C). In the presence of NBQX-APV, the extended responses to the light stimuli disappeared (Figs S4C and S5D). Based on previous observations, the extended phase of the response is derived from the activation of synaptically connected neurons (Wagenaar et al, 2004). Our data demonstrate the development of

synaptically connected functional neuronal networks across the MEA chambers.

Taken together, our long-term MEA electrophysiology data revealed different functional features of developing iNGN networks. Neuronal activity shifted from local and sparse APs to burst activities over time (Figs 2A and S2C, and 2F). The raster plot profiles of electrode and individual neurons showed synchronized network activities (Figs 2C and D and S2A). Optogenetic stimulation also activated initially inactive neurons and led to increased firing rates at earlier developmental time points, suggesting that optogenetic stimulation boosts neuronal activity of hiPSC-derived neurons.

### Precise optogenetic activation of single neurons by holographic stimulation

Whereas full-field optogenetic stimulation resulted in robust neuronal network responses, precise sub-network activities were masked. Combined holographic stimulation with MEA recordings provided high spatial stimulation resolution of 8-$\mu$m spots (Fig S6) (Schmieder et al, 2018). Before holographic stimulation, we captured fluorescence images to visualize ChR2-EYFP–expressing neurons across the electrode grid. We used these coordinates for automatic guidance of holographic stimulation to individual ChR2-expressing cells. A holographic stimulation episode of >20 pulses of 50 ms blue laser light (450 nm, 0.15 W/mm$^2$) at 2 Hz (Fig 3A and C) was presented to each neuron. On average, 400 brightly ChR2-EYFP–labeled neurons per MEA (Fig S7) were individually stimulated. We detected robust light-induced activity by holographic stimulation (Fig 3A and C).

Next, we compared full-field versus holographic optogenetic activation of iNGN neurons within identical MEA cultures (N = 4 MEAs) at different time points. Baseline activity between stimulation pulses was excluded from our analyses. The percentage of electrodes which responded to holographic versus full-field stimulation was similar (Fig 3E). As expected, the average AP frequencies triggered by holographic stimulation were significantly lower than full-field stimulation (P < 0.001; Fig 3F). Holographic stimulation also increased the number of detected neuronal units (Fig 3B and G). Even though, at earlier days (dpi 35–60) the number of identified neurons was slightly lower than full-field stimulation, at dpi 70 and 80 the average number of detected neurons per culture was in the same range (Fig 3G). Neural firing rates remained constant over consecutive episodes of holographic stimulation (Figs 3I and S8), showing that the neuronal activity was not affected by the long-term holographic stimulation experiment. Within all applied holographic stimulation episodes, few episodes triggered a significantly higher response in a neuron (Figs 3H and I and S8). A selective response of each neuron to specific episodes of holographic stimulation was also observed in the long-term raster plot activity profile (Fig 3C). The average neuronal firing rate in the responded episodes (see the Materials and Methods section, Single

(up-left), number of neurons firing in each frequency domains (up-right), burst frequency per neuron (bottom-left), and percentages of APs in burst (bottom-right). All full-field stimulation data presented here were averaged over the 50-ms stimulation period plus 25-ms post-stimulus time across four multi-electrode array cultures (n = 239 neurons). Points represent neurons and big circles show the average data for each day. All data are represented as mean ± SEM. Differences among groups are tested by mixed-effects analysis followed by Sidak's multiple comparisons test (**P < 0.01, ***P < 0.001, ns not significant).

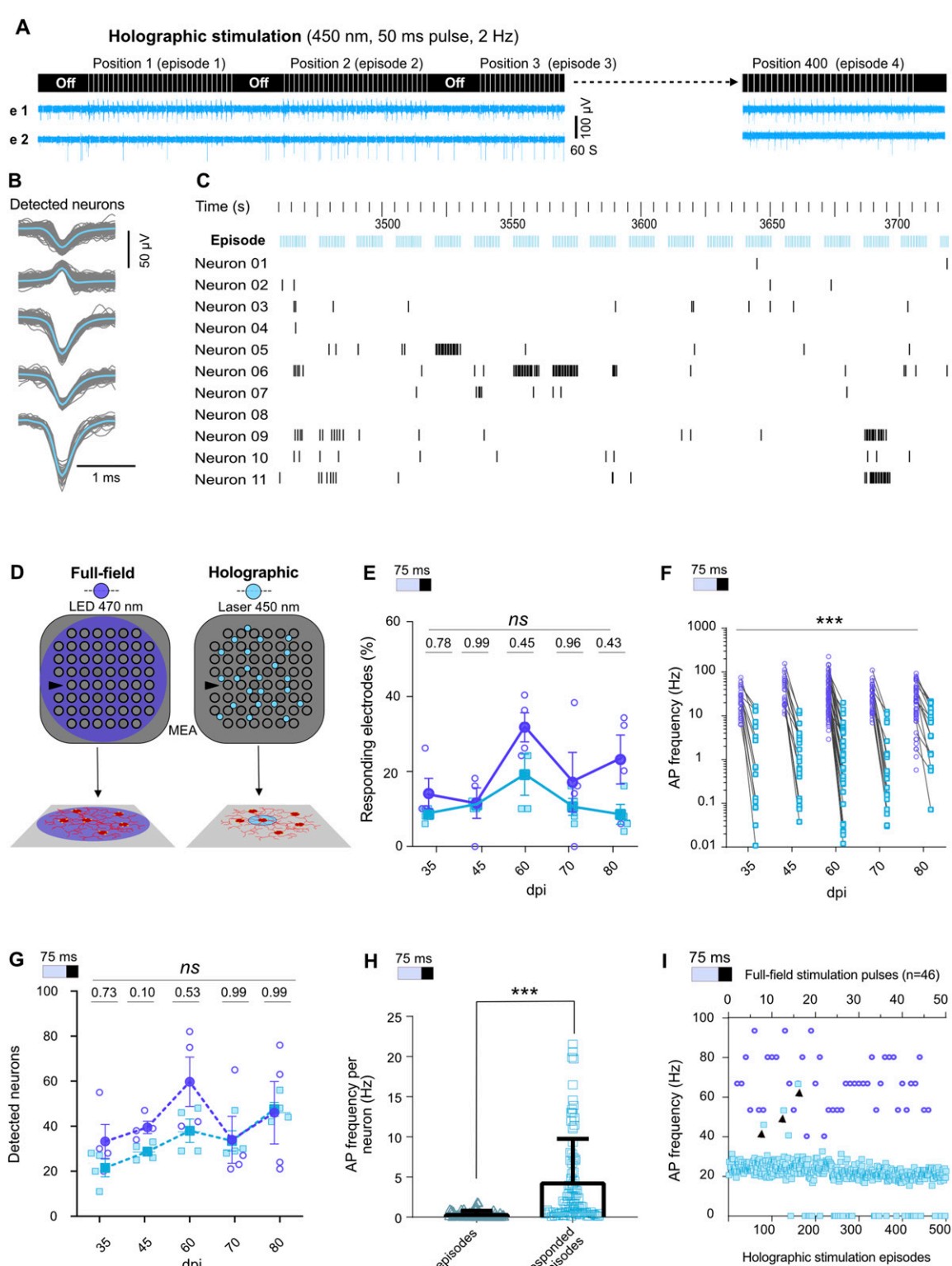

**Figure 3. Activity evoked by holographic (light blue) versus full-field (dark blue) stimulation.**
**(A)** Protocol of holographic stimulation: neurons were detected by fluorescence microscopy and then stimulated 20–30 times during each holographic stimulation episode. Every holographic experiment at each days post induction targeted on average 400 neurons, each stimulated by an episode (upper panel). Sample profiles of activity in two electrodes (e 1 and e 2) show distinct responses to holographic stimulation of neurons at different positions in the network (lower panel). **(B)** Detected neuronal units from holographic stimulation data of the same electrode as in Fig 2E. **(C)** Raster plot of 11 neuronal unit responses to 17 episodes of holographic stimulation. **(D)** Schematic of full-field stimulation affecting the whole network versus holographic stimulation targeting individual neurons. **(E)** Percentage of electrodes

neuron stimulation functional connectivity maps) was 3.64 ± 0.55 Hz (Fig 3H). These episodes were determined by a significant change in temporal spike distribution and validated using a Kolmogorov–Smirnov (KS) test. Maximum AP frequencies evoked by holographic stimulation were in similar ranges as evoked by full-field stimulation (Fig 3I). In general, holographic stimulation resulted in more local activity than full-field stimulation, which is reflected in locally synchronized responses in some neurons (Fig 3C) rather than synchronized activity across all detected neurons (Fig 2D).

### Distance-dependent responses of neurons to holographic stimulation

In random hiPSC-derived neuronal networks, only very few neurons on an MEA are located close to a recording electrode and are therefore accessible for direct recordings. In contrast, holographic stimulation provides spatial control. Thereby, PSTH features with regard to the physical distance between stimulated neurons and recording electrodes can be extracted. We analyzed data from six electrodes of one MEA that were consistently activated by holographic stimulation at different distances between the electrode and the stimulation site (35–80 dpi). Holographic stimulation of nearby neurons, <30 $\mu$m, evoked significantly higher PSTH amplitudes ($P < 0.01$ versus larger distances, Fig 4C). Neurons located <30 $\mu$m also represented 27.79 ± 1.53 ms PSTH peak delay from stimulus onset, which was shorter than PSTH peak delays for more distant neurons ($P < 0.01$, Fig 4D). Based on our data obtained from NBQX-APV–treated networks, direct neuronal responses to optogenetic stimulation were mainly recorded with less than 50-ms delays (Figs S4C and S5D). Correlation analysis between PSTH peak amplitude and its peak delay showed that high PSTH peak amplitudes were attributed with shorter latencies (Fig 4E). Therefore, direct neuronal responses to holographic stimulation were mostly recorded within 30 $\mu$m, whereas lower PSTH amplitudes and larger latencies corresponded to longer distances and reflected indirect responses through synaptically connected functional neuronal networks (Fig 4C–E).

### Measuring connectivity motifs by precise holographic stimulation

We aimed to extract subnetworks within the random neuronal networks by tracking propagating evoked neuronal activity across the entire MEA area. Nearby and distant electrodes which indirectly received propagated neuronal activity were extracted from MEA data by PSTH latency analysis. When we simultaneously used full-field stimulation on all ChR2-expressing neurons, all latency-related information was lost (Fig 5B). In response to the full-field stimulation, different electrodes showed almost identical PSTH peak positions, as each electrode recorded direct neuronal activity at least from one ChR2-activated neuron (Fig 5B). However, by

holographic stimulation of individual neurons, we detected that the PSTH profiles varied between neurons depending on direct or indirect activation by a holographic stimulus (Fig 5A, C, and D). Hence, single-neuron holographic stimulation enabled tracking of propagating activity across sub-networks of neurons based on PSTH timing to reveal connectivity motifs (Fig 5E).

### Tracking the dynamics of connectivity maps over time

Several methods have been proposed for extracting functional connectivity maps from baseline MEA recordings (Bastos & Schoffelen, 2016; Kapucu et al, 2016). For standard MEA recordings, however, many neurons in the network are beyond the typical recording distance of the electrodes and therefore not directly observable (Fig S1). Directly stimulating individual neurons facilitated the extraction of whole-network functional connectivity maps between stimulated and recorded neurons as a superposition of all the connections found (Fig 6A and D). We collected connectivity data from different electrodes (n = 17) that showed a validated stimulus response to at least one of the, on average, 400 target neurons per MEA (N = 4; Fig 6D). For comparison, we extracted electrode–electrode functional connectivity from baseline recordings using spectral entropy (Kapucu et al, 2016) (Fig 6B) and from single-neuron holographic stimulation recordings (Fig 6C). For this representation, the connection strength between two electrodes was measured as the sum of simultaneous responses to one stimulation episode, normalized to the overall number of stimulation episodes. This analysis was crucial to determine the differences and similarities between functional connectivity maps. Connectivity maps from cultures of different ages showed noticeable changes, including shifts in functionally connected electrodes and changes in the number of electrodes connected to each targeted neuron (Figs 6A and S9).

These connectivity maps (N = 4 MEAs) at different ages were used to reveal changes in connectivity parameters over time (Fig 6E–G). The total number of functionally connected neurons to each electrode was counted in electrodes that showed KS-test-validated connections (n = 56) at different days. The average number of neurons connected to each electrode (n = 56 electrodes) increased from 49.18 ± 16.64 at 35 dpi to 217 ± 65.74 at 80 dpi (Fig 6E). The average number of functionally connected electrodes to each target neuron (N = 4 MEAs and n = 432 to n = 868 target neurons depending on day of experiment) increased from 1.65 ± 0.05 at 35 dpi to 4.19 ± 0.11 at 60 dpi ($P < 0.001$), followed by a decrease at 70 dpi ($P < 0.001$ versus 60 dpi), and reached its peak at 80 dpi (5.72 ± 0.05, $P < 0.001$ versus 60 dpi) (Fig 6F). The connection strength measured as the correlation coefficient between the stimulated spike train of spike-sorted APs and the stimulus signal slightly increased with culture age, but this trend was not statistically significant (Fig 6G). Here, the connectivity parameters revealed new aspects of iNGN

responding to full-field or holographic stimulations. Each point represents one multi-electrode array (MEA) (n = 4 MEAs). **(F)** Average activity of neuronal units in each electrode in response to full-field and holographic stimulation as measured over time (N = 4, n = 95). **(G)** Average number of detected neuronal units per MEA (n = 4) at different days. *P*-values are noted above each day. **(H)** Average AP frequency of neurons in successful holographic stimulation episodes versus all applied episodes. **(I)** AP frequencies recorded during 500 subsequent holographic stimulation episodes (blue squares) or 46 pulses of full-field stimulation (blue circles) on 35 days post induction. Black arrows indicate three holographic episodes that induced higher-frequency APs. For (E, F, G), data were compared using mixed-effect analysis followed by Sidak's multiple comparisons test and for (H) Wilcoxon test was applied (***$P < 0.001$).

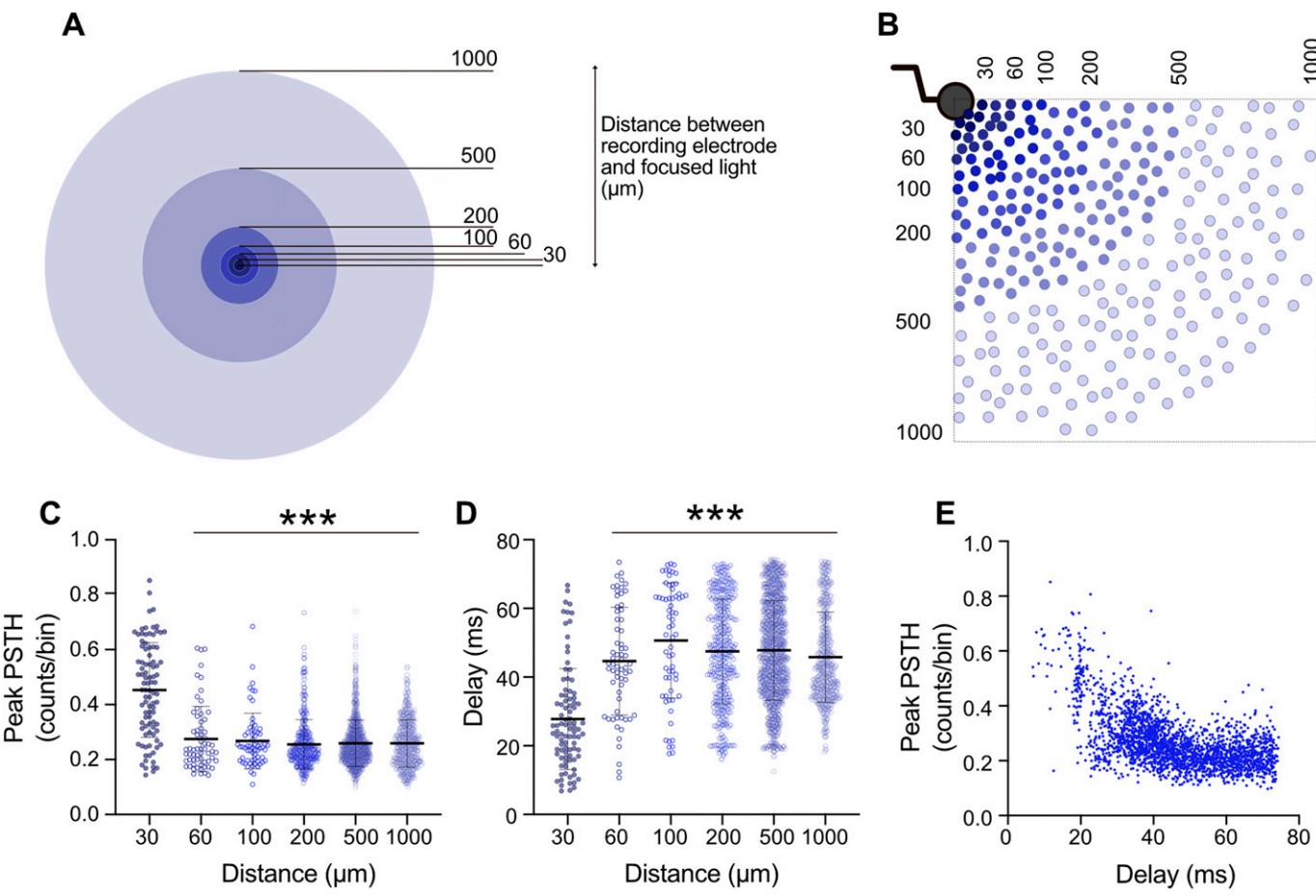

**Figure 4. Distance-dependent direct and indirect responses to holographic stimulation.**
**(A)** Categorization of distances between each holographically stimulated neuron and each of the six analyzed electrodes was based on their radial distances from the edge of the electrode. **(B)** Sample schematic depicting distribution of stimulation points (ChR2-EYFP–expressing neurons) in relation to an electrode. **(C, D)** For each target neuron, 20 pulses were applied (450 nm, 50 ms, 2 Hz) and PSTHs were measured at six active electrodes. PSTH peak value and measured delay from stimulus onset were extracted and plotted based on categorized distances. The number of PSTHs studied at distances of 30, 60, 100, 200, 500, and 1,000 $\mu$m was 105, 67, 65, 379, 1,276, and 880, respectively. **(C, D)** Each dot represents a PSTH peak value (C) or peak position (D) for individual stimulation episodes. Average values represented as mean ± SEM with thick horizontal lines and vertical error bars. Mixed-effect analysis followed by Sidak's multiple comparisons test was applied to compare between different distances. ***$P < 0.001$ versus the mean value at 30 $\mu$m distance. **(E)** Correspondence between individual PSTH peak values (amplitude) and PSTH peak positions (delay). Data were collected from recordings from one multi-electrode array, 35–80 days post induction.

network development such as dynamically changing connection strength independent of network activity levels over time. Tracking multi-parametric network functional features with holographic stimulation extends our understanding of functional aspects of neuronal network development. Comparable to the mapping of functional connections in vivo, for example, to reveal how the functional organization of the brain is altered by neurological disorders (Constable et al, 2013), in vitro mapping of neuronal network connectivity and activity features can be exploited for disease modeling using hiPSC-derived neuronal networks.

## Discussion

To date, most studies on deciphering functional properties of hiPSC-derived neuronal networks have mainly evaluated the basic functional features such as neuronal firing rates and burst features

(Shi et al, 2012; Kuijlaars et al, 2016; Lu et al, 2019; Ronchi et al, 2021). Here, we combined time-lapse functional MEA studies of developing human stem-cell-derived neuronal networks with full-field and holographic optogenetic stimulation platforms, providing access to multiple functional features over time. Key to our study was the establishment of a robust long-term neuronal network culturing system, facilitating stable MEA recordings over months. Our two-step protocol, first inducing neurogenesis on Matrigel-coated surfaces, followed by re-seeding of neuronal precursors on poly-D-lysine (PDL)-coated MEAs, was crucial to provide long-term neuronal adhesion to the MEAs. We further supported the neuronal cultures with specialized commercial media for neuronal maturation and function. In addition, we integrated an established astrocyte co-culture system in which astrocytes were placed in close proximity to, but still separated from, neurons (Fig 1). The beneficial role of astrocytes on hiPSC-derived neurons has been well established resulting in increased firing rates, dendritic

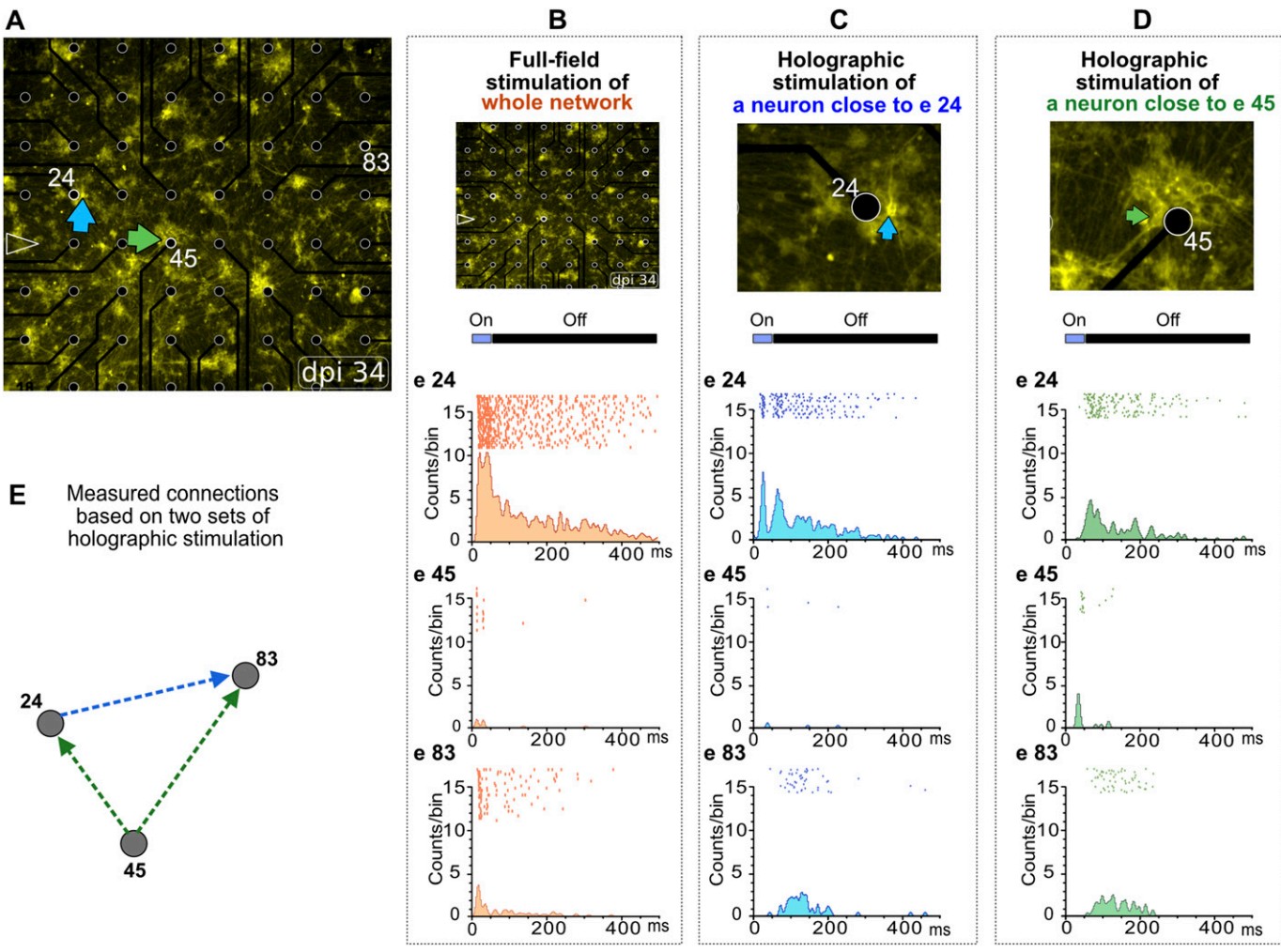

**Figure 5. Representative connectivity patterns based on PSTH profiles of holographic stimulation.**
**(A)** At 35 days post induction, two neurons close to electrodes 24 and 45 (e24 and e45) were stimulated separately by two episodes of holographic stimulation **(B, C, D)** or simultaneously by full-field stimulation (B). Distance between electrodes, 200 $\mu$m. **(B, C, D)** Peri-event raster and histograms of evoked activity in e24, e45, and e83 in response to full-field (B) or holographic (C, D) stimulation. **(C)** Holographic episode targeting a neuron close to e24 (C) induced a direct response with short delay at e24 and delayed response at e83 but did not affect the activity at e45. **(D)** Another episode targeted a neuron close to e45 (D) that evoked a direct response with short delay at e45, and indirect responses with long delay at e24 and e83. **(E)** Based on collective PSTH profiles of e24, e45, and e83 in response to the holographic stimulation of two individual neurons, a functional connectivity motif within this random neuronal subnetwork was extracted.

arborizations, and maturation of excitatory glutamatergic synapses (Klapper et al, 2017). Media evaporation leading to pH and osmolarity changes were another challenge for long-term electrophysiology recordings outside of the incubator (Biffi et al, 2013; Saalfrank et al, 2015). Custom made PDMS lids prevented media evaporation while allowing gas exchange (Blau et al, 2009), which facilitated long holographic stimulation sessions of the same MEA preparation over the time course of functional network development.

Our recorded spontaneous activity data are in line with previous studies using hiPSCs in which neurogenesis was triggered by neurogenin-1 and -2 overexpression (Lu et al, 2019). Sparse APs were detected from 25 dpi onwards, and developed into burst activity, reaching a peak around 60 dpi followed by a slight decrease in activity at later time points. Developing primary cortical and hippocampal networks have shown comparable trends in activity

over time in vitro, but these cells were already active at 14 days in vitro (div) and a peak activity was detected at 21 div, followed by decreased activity levels at later time points (Chiappalone et al, 2006, 2007; Habibey et al, 2015). Hence, the development of iNGNs, and also other human stem-cell–derived neurons (Hyvärinen et al, 2019), very likely reflect similar maturation patterns regarding neuronal activity. However, human neuronal cultures require longer time periods to become fully functional. Previous studies based on whole-cell patch-clamp recordings from iNGN neurons at different developmental time points showed a trending decrease in resting membrane potential (RMP) up to 35 dpi (–75 mV) (Lam et al, 2017). The decrease in RMP was concomitant with increased expression of voltage-gated sodium and potassium channels and elevated spontaneous postsynaptic currents (sPSCs) (Busskamp et al, 2014; Lam et al, 2017). Whereas previous patch-clamp studies required different neuronal preparations for each recording day,

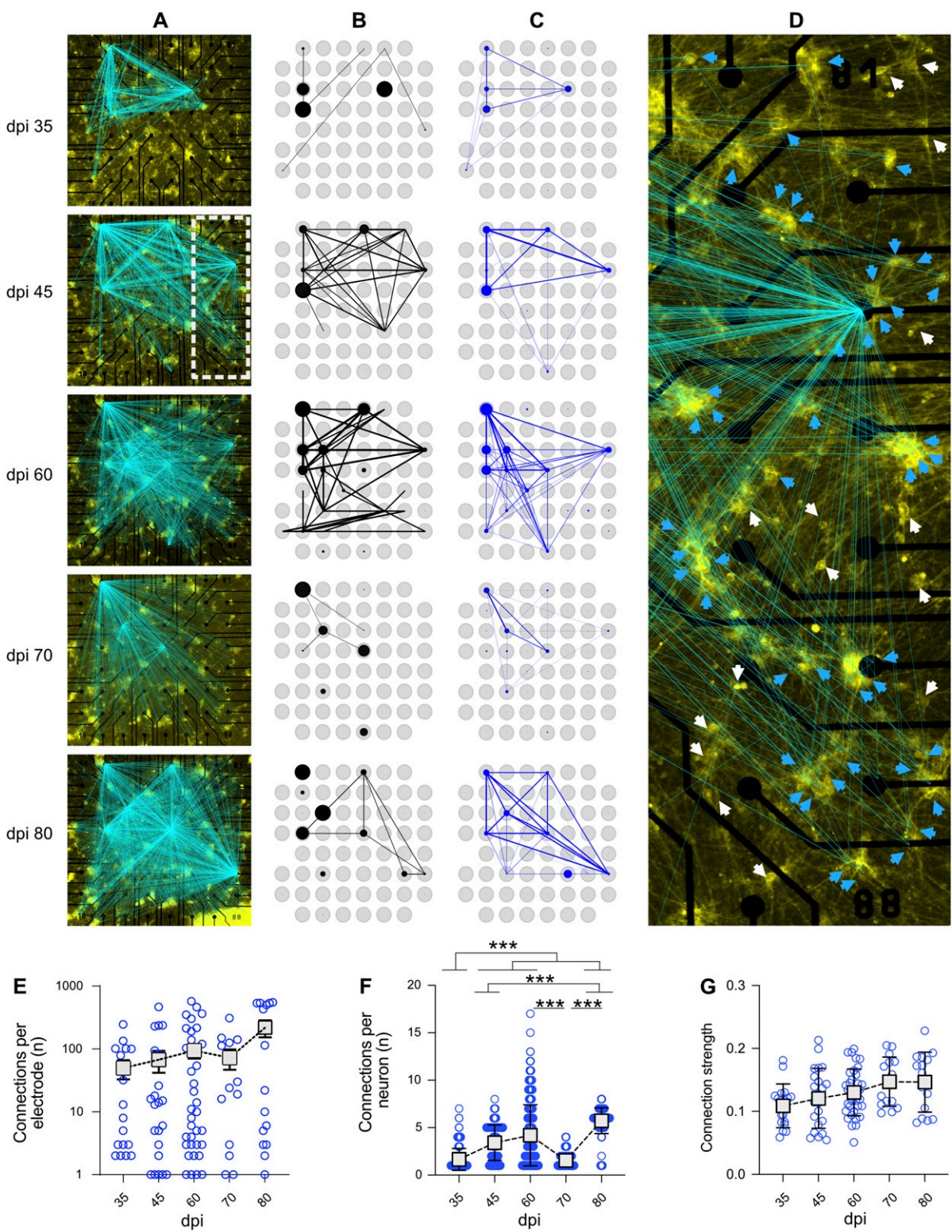

**Figure 6. Functional connectivity maps over time based on spike-clustered data.**
**(A)** Development of the number of validated connections between stimulated neurons and recording electrodes over 80 days post induction. Lines indicate a valid connection and are overlaid on the fluorescence image of the sample on the day of recording. Line intensity scales with connection strength. **(B, C)** Functional connectivity of the sample calculated for baseline activity and (C) holographic stimulation. Thickness and transparency scale with connection strength. Blue and black circles show general activity of the respective electrode, scaled to the maximum for each mode of experiment. **(A, D)** Shows a magnified view of the sample at 45 days post induction (marked as white rectangle in (A)). White arrows indicate neurons which were stimulated but did not show a significant response, whereas blue arrows

our MEA study on identical cultures over time corroborated all findings on in vitro human neuronal network maturation. Burst activity is mainly derived from contributions of excitatory and inhibitory synapses (Chiappalone et al, 2007; Suresh et al, 2016). Spontaneous burst activity in iNGN networks appeared at around 35 dpi and reached its peak at 60 dpi. In hiPSC-derived brain organoids, highly synchronous activity has also been reported at around day 60: this later changes to rhythmic activity (Trujillo et al, 2019).

Instead of just waiting until spontaneous neuronal activity started to occur, we applied optogenetic stimulation. Indeed, we detected robust neuronal activity at earlier time points, which is in line with previous studies using electrical stimulation for hiPSC-derived (Amin et al, 2016; Cho et al, 2016) and optogenetic activation of mouse ESC-derived neurons (Stroh et al, 2011). We further showed that full-field stimulation evoked a combination of direct and delayed PSTH peaks, highlighting synaptically connected functional neuronal networks that were blocked by AMPA- and NMDA-receptor antagonists (Figs S4 and S5). By engaging more neurons to network activity (Fig 2E) and increasing firing rates (Fig 2F), optogenetic stimulation serves as a booster for hiPSC-derived network maturation (Stroh et al, 2011; Yu et al, 2019). Still, further functional studies are required to explore the optimal continuous optogenetic stimulation parameters to accelerate functional maturation of human stem-cell-derived neurons. In our study, the neuronal cultures were stimulated by light during the recording sessions. Testing continuous and regular light stimulation within the incubators might further accelerate neuronal network maturation. This intervention will significantly decrease the human neuronal culturing period, saving valuable resources.

Targeting individual neurons by high spatiotemporal holographic optogenetic stimulation systems has been applied for brain tissue and slices to extract connectivity maps (Ronzitti et al, 2017; Chen et al, 2018). For example, holographic stimulation has been applied to stimulate the high-photocurrent channelrhodopsin CoChR. This optogenetic actuator was localized to the soma of neurons within cortical slices and exploited for tracking the functional connectivity maps between pre- and post-synaptic neurons (Shemesh et al, 2017). Our holographic stimulation platform revealed functional subunits within neuronal networks, which was not accessible based on full-field stimulation data. When all ChR2-expressing neurons were activated simultaneously by full-field stimulation, connectivity patterns between neurons were mostly masked. Holographic stimulation of ChR2-expressing neurons enabled single-cell activation, which allowed us to extract the propagating neuronal activity within sub-networks. For semi-automated holographic stimulation, on average 400 neurons were marked manually, followed by automated stimulation using a programmable pattern. As a proof of principle, repeated stimulation sessions at different culture ages provided insights into connectivity motifs that showed dynamic connectivity patterns: the

total number and strength of functional connections increased with culture age and was dependent on the network activity levels.

So far, connectivity studies have only been performed on developing primary cortical and hippocampal circuits using spontaneous activity data (Garofalo et al, 2009; Ullo et al, 2014; Poli et al, 2015) and with hiPSC-derived networks based on electrode-to-electrode functional connections (Hyvärinen et al, 2019). However, the latter method excluded a major part of the network activity, as most neurons were located between electrodes and therefore not accessible to be stimulated (Nam & Wheeler, 2011). Here, we demonstrated that holographic stimulation enabled correlating electrode activity to neurons located far from electrodes, and thus increased the spatial resolution of the connectivity maps. The low spatial resolution of the standard MEA chips with 200-$\mu$m electrode pitch could be overcome using cMOS-based MEAs that offer higher spatial resolution (17-$\mu$m electrode pitch). Therefore, integrating the holographic optogenetic stimulation with cMOS-based recording could be exploited to improve the resolution of the extracted functional connectivity maps. However, cMOS-based MEAs are not transparent, which makes it challenging to provide high-resolution live images from the network and to spot neurons individually. On the other hand, cMOS-based systems also offer electrical stimulation with high-spatial resolution that might be a substitute for focalized holographic stimulation. In addition to the stimulation of excitatory neurons, one can also use holographic illumination to exploit inhibitory optogenetic actuators for silencing neurons. The field of view (FOV) of our holographic illumination system is defined by the optical system and limited to an area of 1.5 × 1.5 mm$^2$ in the configuration presented here. Omitting the magnification optics, an FOV of around 10 × 6 mm$^2$ and a spatial resolution of 15 $\mu$m has been achieved (Schmieder et al, 2018). Increasing the magnification, on the other hand, will allow for subcellular stimulation experiments at a smaller FOV. Altogether, holographic optogenetic stimulation promises higher flexibility on studying functional network features than cMOS MEAs alone. Future experiments may combine strengths of both techniques. Furthermore, single neuron stimulation in thick samples could be achieved by combining our setup with two-photon holographic stimulation methods (Paluch-Siegler et al, 2015; Ronzitti et al, 2017; Gill et al, 2020).

It has been shown that illumination with blue light in combination with standard BrainPhys medium can result in cytotoxic effects due to an increase of $H_2O_2$ in the cell culture medium (Zabolocki et al, 2020). In this case, severe cytotoxicity was observed after 24 h of ambient light and blue LED illumination. In our study, even though the individual experiments exceeded 1 h, using intervals of 50 ms the overall illumination time was just 1/20$^{th}$ of that. Furthermore, for holographic stimulation, the light pulses were only applied on individual neurons rather than illuminating the entire network. Tracking neuronal activity across all applied episodes

indicate neurons for which stimulation elicited a significant response. **(E)** The average number of neurons connected to an active electrode, normalized to the overall number of stimulated neurons per experiment (N = 4 multi-electrode arrays [MEAs] and n = 56 electrodes with KS-validated connections). Connections per electrode represented in logarithmic scale (log 10) to visualize all data range. **(F)** Number of neurons showing a valid response to the stimulation of a target neuron on each MEA (N = 4 MEAs). **(G)** Strength of the connection of a neuron to the target neuron (N = 4 MEAs and n = 56 electrodes). Connection strength is measured as the maximum of the cross-correlation of the stimulus response of one neuron with the stimulus, represented by a train of boxcar functions. Data were compared between days using Kruskal–Wallis test followed by Dunn's multiple comparisons test. *$P$ < 0.05, **$P$ < 0.01 and ***$P$ < 0.001 versus specified date. Distance between electrodes, 200 $\mu$m.

showed a stable firing rate in all electrodes (Figs 3I and S8), suggesting that phototoxicity did not influence our experiments.

## Conclusion

Overall, we demonstrated that holographic optogenetic stimulation of human neurons is robust and, regarding precise functional properties; superior to full-field stimulation. Depending on the particular experiment, our holographic stimulation platform is also capable of targeting several neurons simultaneously by generating multiple foci (Schmieder et al, 2018). Furthermore, the flexibility of our system will be further improved by integrating combinations of the optogenetic actuators and controlling different neurons with diverse wavelengths. These technological advances have extended the competence of the holographic stimulation for deciphering precise functional features of developing hiPSC-derived networks: this will serve both basic biomedical research as well as studying pathophysiological features of developing networks derived from patient hiPSCs. Specifically, these hiPSC-derived neurons are easy to adapt to particular research questions. Here, we provide a methodological entry point to capture high-resolution functional data of individual neurons organized in sub-circuits within entire random networks over time. These data provide an extended view on precise functional differences. Having fully functional human neurons and single-cellular activation control are important steps to guide this model system out of its infancy and enable it to become accepted as a complementary platform to animal research within the neuroscience community.

# Materials and Methods

### Long-term hiPSC-derived neuronal culture on MEA chips

We adapted a two-step cell-seeding protocol that included short-term induction of iNGN cells for 5 d in a Matrigel (Corning)-coated well plate followed by re-seeding differentiated neurons on PDL (Merck)-laminin (Sigma-Aldrich)-coated MEAs (Multi Channel Systems). Human-derived iNGN neurons (Busskamp et al, 2014) were thawed and passaged twice before seeding them in Matrigel-coated well plates. These cells were cultured in mTeSR1 medium (mTeSR1 Basal Medium with mTeSR1 5× Supplement [Stemcell Technologies] and 1% penicillin–streptomycin [P/S; Thermo Fisher Scientific]). For neural induction, iNGN cells were treated with 0.5 μg/ml doxycycline (Sigma-Aldrich) for 5 consecutive days (Fig 1). To initiate the expression of EYFP-tagged ChR2 in iNGN cells, a lentiviral vector (plv-ef1a-ChR2-EYFP) was added 1 d after seeding within a biosafety level-2 facility. Undifferentiated cells were removed by adding Ara-C (5 μM final concentration; cytosine β-D-arabinofuranoside hydrochloride; Sigma-Aldrich) at day 4 post induction (4 dpi). Standard MEA chips were plasma treated for 2 min (ambient air, 0.3 mbar), coated with PDL (50 μl per electrode area in 0.1 mg/ml final concentration) and incubated overnight at 37°C. MEAs were washed with sterile deionized water (diH$_2$O) (three times, each time 10 min) and dried under the hood with laminar flow. Laminin (50 μl; 0.05 mg/ml concentration) was added to the

PDL-coated MEAs and incubated for 3 h before reseeding the 5 dpi iNGN cells, which were dissociated by incubating them with Accutase (Sigma-Aldrich) for 5 min. The dissociated cell suspension was centrifuged (359g, 4 min), supernatant was removed and re-suspended in complete BrainPhys (BrainPhys Neuronal Medium [Stemcell Technologies] + 1% P/S + NeuroCult SM1 Neuronal Supplement [Stemcell Technologies] + N2 Supplement-A + 20 ng/ml recombinant human BDNF [Peprotech] + 20 ng/ml recombinant human GDNF [Peprotech] and 200 nM ascorbic acid [Sigma-Aldrich]). Differentiated cells were re-seeded on PDL-laminin–coated MEA chips (around 100,000 cells per cm$^2$ including electrode area).

Rat primary astrocyte cultures (A1261301; Thermo Fisher Scientific) were prepared in parallel (Fig 1) on PDL- and laminin-coated coverslips (0.1 and 0.05 mg/ml, respectively). Paraffin dots were already placed on the coated side of the coverslips (Fig 1). Astrocytes were cultured in astrocyte medium including DMEM + 4.5 g/l d-glucose + pyruvate plus N2 Supplement, 10% One Shot fetal bovine serum and 1% P/S (all provided by Thermo Fisher Scientific). Astrocyte medium was changed to complete BrainPhys media 1 d before re-seeding the neurons on MEA chips. Astrocyte cultures on paraffin coverslips were washed with PBS without calcium and magnesium (Thermo Fisher Scientific), flipped and placed on top of the re-seeded neural cells. Then, complete BrainPhys (BrainPhys Neuronal Medium [Stemcell Technologies] + 1% P/S + Neuro-Cult SM1 Neuronal Supplement [Stemcell Technologies] + N2 Supplement-A + 20 ng/ml recombinant human BDNF [Peprotech] + 20 ng/ml recombinant human GDNF [Peprotech] + 200 nM ascorbic acid [Sigma-Aldrich]) was added to the MEA rings, covering neuronal and astrocyte cultures (Fig 1). Half of the medium was exchanged with fresh complete BrainPhys every week.

### Microscopy

An inverted EVOS FL Imaging System (Thermo Fisher Scientific) was used to capture brightfield (10× and 20×) and fluorescent images (20×) from neuronal cultures on MEA chips at different days post induction (Fig 1). High-resolution fluorescent images were used to detect neural cells expressing the EYFP-tagged ChR2. In all full-field and holographic light-stimulation experiments, high-resolution images were prepared from all cultures just before MEA recording. From each MEA culture, 20 images (20× magnification) were acquired to cover the entire electrode area (1.5 mm$^2$). These images were then automatically stitched using ImageJ software (Preibisch et al, 2009), adjusted for their contrast, and used to target individual neurons in holographic light stimulation (Figs S1 and S7).

### MEA electrophysiology

Neural network spontaneous and light-evoked activities were recorded using standard MEA chips (60 MEA200/30iR-Ti-gr) on an MEA1060-Inv-BC amplifier (sampling rate 25 kHz) using the MC_Rack software, all provided by Multi Channel Systems (MCS). In each MEA chip, 59 electrodes (8 × 8 matrix of electrodes including a counter electrode) simultaneously recorded extracellular activities from different parts of the network. Data acquisition was performed at a sampling frequency of 25 kHz. During the recording, temperature

was controlled by a built-in temperature controller (TC02, MCS). The MEA chip and amplifier were mounted on an inverted microscope (Nikon Eclipse Ti). All recordings were performed in a dark Faraday cage shielding external electromagnetic and optical noises. Spontaneous activity was recorded from 15 dpi onward (Fig 1). Baseline spontaneous activity was recorded for 3 min before starting the full-field stimulation experiment on all recording days. Recording of spontaneous and evoked activities was performed in the presence of astrocyte culture.

## Full-field optogenetic stimulation

Before performing functional recordings, the center of each MEA chip (electrode area) was aligned with the center of the 640-nm light beam and its focal plane. Full-field stimulation and functional recordings were performed from 15 dpi (Fig 1). A spectra 4 (Lumencor) LED light source was used to apply full-field stimulation (470 nm, 50 ms pulses, 0.5 Hz) at 0.46 mW/mm$^2$ through a 4× objective lens of the inverted Nikon microscope. Blue light pulses were applied without including the filter cubes in the light path. The intensity of the light pulses was adjusted using commercially available software (Lumencor Graphical User Interface). Stimulation protocols including pulse width, intervals (frequency), and number were controlled by a custom Python script and pCLAMP 11 software (Molecular Devices). Full-field stimulation light pulse properties including light-ON and light-OFF timestamps were digitized and recorded parallel to electrophysiology signals in MC-Rack recording software (MCS).

## Inhibition of excitatory and inhibitory synapses

Two MEA cultures were used to block the excitatory glutamate receptors in the network at 110 dpi. We applied a combination of 2,3-dihydroxy-6-nitro-7-sulfamoyl-benzo[f]quinoxaline (NBQX; an AMPA receptor antagonist) and (2R)-amino-5-phosphonopentanoate (APV; a selective NMDA receptor antagonist). Before NBQX+APV treatment, spontaneous activity and response to full-field stimulation was recorded. Half of the medium in the MEA ring was drained and kept in an incubator. Then NBQX (10 $\mu$M final concentration) and APV (50 $\mu$M final concentration) were prepared in 500 $\mu$l of fresh BrainPhys medium and added to the rest of the medium in the MEA ring. After 5-min incubation, spontaneous activity was recorded in the presence of the NBQX+APV, and then full-field stimulation was applied at different light intensities (Figs S4, S5, and S10). Later, NBQX+APV was washed out two times with warm complete BrainPhys medium. Then the pre-used medium plus fresh medium were mixed (1:1) and added to the MEA chamber. After 30 min of incubation, the activity was recorded again under non-stimulated and full-field stimulated conditions.

## Holographic stimulation

### Holographic stimulation setup
The setup used has been reported previously and can be seen in Fig S11 (Schmieder et al, 2018). Collimated light from a laser diode (450 nm; Thorlabs) illuminates a ferroelectric liquid crystal on a silicon SLM (QXGA-R9, 1,536 × 2,048 pixels, 8.2 $\mu$m pixel pitch; Forth

Dimension Displays), which offers binary phase modulation at a maximum refresh rate of 400 Hz. Light reflected off the SLM is converted to true binary phase modulation using a linear polarizer. Fresnel Holograms displayed by the SLM are demagnified by a Keplerian telescope (magnification $\beta$ = 25 mm/180 mm = 1/7.2). The minimum focusing distance for our setup according to Schmieder et al (2018) is $f_{min} = \beta \frac{N \Delta r^2}{\lambda}$. With N = 1,536 being the number of pixels along the shorter side of the SLM, $\Delta r$ = 8.2 $\mu$m being the pixel pitch of the modulator, and $\lambda$ = 450 nm the wavelength of the laser, $f_{min} \approx$ 6.4 mm for this setup. Taking into account the focal length $f_3$ of the lens facing the sample, the overall working distance is then $d_w = f_{min} + f_3$ = 31.4 mm, which is required to fit the MEA recording device underneath the front optical element. The minimum focus diameter achieved was 8.1 $\mu$m. As an improvement over (Schmieder et al, 2018), the system is now calibrated and operated using a self-programmed graphical user interface (GUI) written in MATLAB. Co-planarity of the sample and the focal plane of the stimulation system was ensured by matching maximum contrast of MEA electrodes with a minimal focus diameter as observed by the microscope camera. Calibration of the setup is performed by projecting a user-defined set of foci onto the sample. The centers of these foci are then detected using a Gaussian fit function. Sample-plane coordinates of the foci are automatically matched to the SLM-plane coordinates of the Fresnel zone plate centers, using a second order polynomial fit to enable precise stimulation of individual neurons. To match coordinates of the high-resolution stitched fluorescence images to the lower resolution camera images of the MEA system, the electrode grid was used to correlate both images. Centers of mass of the electrode heads of both images are matched using a bilinear image transformation, which accounts for rotation, shift, and linear scaling. This dual calibration makes it possible to calculate holograms which stimulate individual neurons in MEA camera coordinates which were previously selected in high-resolution fluorescence images.

### Hologram calculation
Fresnel holograms were used for holographic stimulation. They were calculated by convolving the desired illumination pattern on the sample with a pre-calculated Fresnel zone plate for the minimum focus distance $f_{min}$, followed by either phase thresholding or bidirectional error diffusion of the phase for binarization (Tsang & Poon, 2013), depending on the complexity of the hologram.

### Holographic stimulation protocol
Before the experiments, individual neurons were manually selected for holographic stimulation from fluorescence images using our self-programmed MATLAB GUI (Fig S7). The GUI was also used to define the stimulation parameters including light pulse duration, frequency, and repetitions. After marking and saving the parameters, holographic stimulation was automatically guided from one targeted neuron to another until all marked neurons were stimulated. Each neuron received 20–30 pulses of 50 ms blue light (450 nm) at 2 Hz, which we call an episode of holographic stimulation (Fig 3). Depending on the network on each MEA and the number of available neurons, on average we applied around 400 episodes, each including one neuron at a particular time point (dpi).

### Data analysis

#### AP frequency and burst frequency at baseline (spontaneous activity)

Long-term activity data were collected from four MEA chips with a total number of 95 active electrodes on seven different days (between 15 and 80 dpi). Recorded electrophysiology data were processed offline. Raw signals were filtered (Butterworth second order, high-pass filter cut-off at 100 Hz) to remove low-frequency noise. The timestamps of the action potentials (APs) were detected by negative and positive thresholds ($\pm 5\sigma$ of the peak-to-peak noise). These timestamps were imported into Neuroexplorer software (Plexon Inc.) to extract AP frequency and burst features. For long-term spontaneous activity we included only active electrodes in the analysis. An electrode is considered active if it showed activity on at least two sessions (days) of recording and the AP frequency was more than 0.2 Hz in both sessions. Most active electrodes showed consistent activity during the whole experimental period. For the electrodes that showed activity only on some days the activity on all other days was set to zero: it was then possible to track the development of activity features over time.

Bursts were detected and extracted using Neuroexplorer software based on the following parameters: 20 ms maximum inter-spike interval to start the burst, 10 ms maximum inter-spike interval to end the burst, 10 ms minimum inter-burst interval, 20 ms minimum burst duration with at least four spikes in each burst (Habibey et al, 2015). Different features of burst activity, including burst frequency, burst duration, and percentage of APs forming burst activity, were extracted (Habibey et al, 2015, 2017; Wilk et al, 2016).

#### AP frequency and burst frequency evoked by full-field stimulation

We applied an identical method of data extraction for collecting the full-field evoked responses from 95 active electrodes of 4 MEAs. Spontaneous activity and full-field evoked activities were compared based on electrode data (Fig S2) and data obtained from sorted neuronal units (Fig 2). Evoked activity in electrodes was extracted between the first and last applied pulse of the full-field stimuli, including both the 50-ms pulse duration and 1,950-ms pulse intervals (light-OFF periods between pulses; Fig S2). To compare the activity in neuronal units, we extracted the APs during 50-ms pulse duration and 25 ms after (75 ms in total). The APs and burst frequencies collected on each recording day were compared with baseline spontaneous activity using mixed-effects analysis of variance followed by Sidak's multiple comparisons test. The number of detected units per MEA was extracted in different days between 35 and 80 dpi and compared between spontaneous and evoked activities using Wilcoxon test.

Baseline AP frequency and burst frequency between NBQX+APV-treated and untreated conditions were also compared using Wilcoxon matched-pairs signed rank test (two MEAs and 39 active electrodes). Light-evoked AP and burst frequencies were compared between NBQX+APV-treated and untreated conditions using the nonparametric Friedman test followed by Dunn's multiple comparisons test (Figs S4 and S5).

#### Comparing activity profiles of full-field versus holographic stimulation

To compare the evoked activity profile resulting from holographic and full-field stimulation, only APs that appeared during the 50-ms light pulses and 25 ms after (75 ms in total) were used. This range included all direct neuronal responses to the light stimulus (Fig 3). Each response to each full-field stimulation pulse represented one data point. For holographic stimulation, average response to 20–30 pulses of each episode represented one data point. Because we applied around 400 holographic stimulation episodes (Figs 3 and S8) each day, around 400 data points were collected for each electrode (Fig 3). The percentage of responding electrodes, number of detected units per MEA, and average activity of neuronal units in each electrode was compared between full-field and holographic stimulations at different days base on mixed-effect analysis followed by Sidak's multiple comparisons test (Fig 3E–G). Significant responses of each neuron to a specific episode of holographic stimulation were identified based on changes in the temporal distribution of spikes based on the displaced impulses function and a Kolmogorov–Smirnov test (see Single neuron stimulation functional connectivity maps for further details). Activity in these active episodes was compared with the responses to all applied episodes together using the Wilcoxon test.

#### Peri-event raster and time histogram

Peri-event raster and time histograms were created by counting the number of APs after the light stimulus (0–500 ms for Fig 4 and 0–200 ms for Figs S4 and S5) and sorting them in 3-ms time bins. Sequences of the responses for repetitive light pulses of each holographic stimulation episode or 23 pulses of full-field stimulation were aligned and superimposed in time.

#### PSTH for holographic stimulation

For the single-cell stimulation data presented in Figs 4 and 6, we calculated PSTHs using self-programmed MATLAB scripts. All spikes occurring in a 75-ms window after the stimulation onset were extracted and sorted into 40-μs bins. The resulting distribution was smoothed by a convolution with a Gaussian function with a width of 3 ms as detailed in the equation below.

$$PSTH(t) = \left[ \left( \sum_{i=1}^{N} \delta(t - t_i) \right) * e^{-\frac{1}{2}\left(\frac{t}{\sigma}\right)^2} \right] \Big/ M$$

with $N$ being the total number of spikes of one stimulation pulse occurring in a 75-ms window after stimulus onset, $t_i$ the arrival times of the individual spikes, $2\sigma = 3$ ms the full 1/$e$ width of the Gaussian function, $M$ the number of stimulus repetitions per episode, and $*$ the convolution operator.

PSTHs of valid connections (see the Materials and Methods section: Single neuron stimulation functional connectivity maps) were only considered if they consisted of more than 20 individual spikes. PSTH maxima were normalized to the number $M$ of stimulations per episode.

Distances between stimulation site and recording electrode were calculated as the Euclidean distance between predefined stimulus location from the fluorescence microscope images and the centers of the recording electrodes.

### Distance-dependent responses of neurons to holographic stimulation

Based on PSTH data for each episode, Euclidean distances between the targeted neuron and six active electrodes of one MEA in one MEA were calculated to correlate response features to physical distances over the entire MEA area (Fig 4A and B). Distances were sorted into several intervals (Fig 4A and B). For each holographic stimulation episode, the frequency of the PSTH peak at each electrode determined the evoked activity level. These data were pooled from all applied holographic episodes at different ages of the same network (n = 1,659 episodes). The latency of the PSTH primary peak from stimulus onset determined the time an evoked response of a neuron required to reach the active recording electrode.

### Baseline functional connectivity maps

Baseline connectivity maps were extracted using the spectral entropy synchronicity measure (Kapucu et al, 2016). Since spectral entropy uses the complete filtered electrode data and not only action potential arrival times, only signals with at least one event exceeding the threshold of 6σ peak-to-peak noise were chosen for further processing to avoid evaluating inactive electrodes. Electrode recordings were split into overlapping segments of equal length. The power spectrum of segments was then normalized to the overall power of the segments, and summed into one value $S_i$ in the spectral entropy signal according to:

$$S_i = \left[ \sum_f (P_{norm}(f) \cdot log(1 \div P_{norm}(f))) \right] \Big/ log(N),$$

with $P_{norm}(f)$ being the normalized frequency power spectrum of segment $i$ with a length of $N$ samples.

The cross-covariance of the spectral entropies of two signals $S_{x,i}$ and $S_{y,i}$ at lag 0 is then calculated as:

$$C_{xy} = \frac{1}{O} \sum_{i=1}^{O} \left( \left( S_{x,i} - \overline{S_x} \right) \left( S_{y,i} - \overline{S_y} \right) \right),$$

with $O$ the number of segments per signal. The cross-correlation coefficient at lag 0 is then:

$$r_{xy} = \frac{C_{xy}}{\sigma_x \sigma_y},$$

with $\sigma_{x,y}$ being the SD of $S_x$ and $S_y$, respectively.

For our experiments, we chose a segment length of 250 ms and an overlap of 80%. To avoid evaluating inactive electrodes, only signals with at least one event exceeding the threshold of 6 σ peak-to-peak noise were chosen for further processing. Since $r_{xy}$ is continuously distributed between –1 and 1, with many values close to 0, a threshold for the most important connections must be defined. In Kapucu et al (2016), this value was picked manually. For displaying the synchronous connections, we tested different thresholds ranging from 0.05 to 0.25. For Fig 6, only connections with a correlation coefficient >0.1 are displayed. Because spectral entropy synchronicity is not calculated from event data, shuffling of data to exclude spurious correlations was executed on $S_x$ before calculating cross correlations. All correlations reduced greatly to values <0.05, well below our threshold for good correlations set for display.

### Spike sorting

For spike sorting, we used the open-source toolbox wave_clus3 (Chaure et al, 2018) for automatic nonparametric spike detection and sorting. For spike detection, raw signals were filtered (Butterworth second order, high-pass filter cut-off at 100 Hz) to remove low-frequency noise. The timestamps of the APs were detected by negative and positive thresholds (±5σ of the peak-to-peak noise). Data from baseline, full-field stimulation, and holographic stimulation experiments were pooled for each MEA and day, evaluated together, and split again in post-processing. Wave_clus3 performs a multiresolution decomposition of waveforms into sets of Haar wavelets, whose scale factors serve as parameters for clustering. Thresholds for the $m$ most significant parameters are chosen automatically. Exceeding the threshold of 40,000, additional spikes are assigned to found clusters by template matching. The clustering method of Chaure et al (2018) is called superparamagnetic clustering. Here, each waveform represents one point in $m$-dimensional phase space with an interaction strength to neighboring points depending on the Euclidean distance between points. From an initially random state of each point, a new random state is assigned to individual points, changing the state of nearby points depending on their interaction strength and the hyperparameter of a virtual temperature assigned to the system. Lowering the temperature, more points will fall into the same state, resulting in clusters of similar waveforms. In Chaure et al (2018), a lower false positive rate and lower overall error rate than comparable unsupervised spike sorting algorithms were reported for superparamagnetic clustering.

### Single neuron stimulation functional connectivity maps

For the evaluation of connectivity using holographic single-cell stimulation, all episodes were evaluated separately. Detected spikes were sorted into separate units by spike-sorting using wave_clus3 (Chaure et al, 2018), a freely available toolset for automatic nonparametric spike sorting based on a wavelet decomposition of spike waveforms. To achieve consistent spike cluster detection, baseline and stimulation experiments were evaluated together and split again in post processing. To detect connections from stimulated to recorded neurons, deviations between stimulated and unstimulated states of single units were distinguished using the displaced impulses function (DIF) (Awiszus, 1993). DIF functions were tested for deviations from baseline using a Kolmogorov–Smirnov (KS) test. Values of $P < 0.01$ were considered to be significantly influenced by stimulation, and therefore a valid connection to the recording electrode. To avoid the possibility of misinterpreting the results based on one burst in the time series, displaced impulses functions were calculated and tested with the KS test for all $N_{stim}$ stimuli of one episode, excluding spikes belonging to a specific stimulus event, resulting in $N_{stim}$ individual tests for one episode. DIFs from electrodes without baseline activity (spikes only evoked by light stimulation) were tested against distribution functions with all values equal to zero. Only episodes passing these $N_{stim}$ tests were finally considered to be a valid connection between the light-stimulated neuron and one unit at a recording electrode. We call these functional connections *valid connections* throughout this article.

Connection strengths were calculated as follows
For a stimulus pulse train,

$$S = \left[ \sum_{k}^{N_{stim}} \delta(t - (\tau_k + 25ms)) \right] \times P,$$

with the single pulse signal $P$,

$$P(t) = \left\{ \begin{array}{ll} 1 & -25ms \leq t \leq 25ms \\ 0 & else \end{array} \right\},$$

where $\tau_k$ are the times of the stimulus onsets, and the connections strength is regarded as the maximum of the cross correlation of $S$ and the spike events $E$:

$$E = \sum_{n=1}^{N_{spike}} \delta(t - t_n),$$

with $N_{spike}$ being the number of spikes in the time interval between the beginning of the first stimulus and the end of the last stimulus of one episode. For better overlap of $S$, $E$ is additionally convoluted with a Gaussian function with a full 1/e width of 3 ms. However, because the overlap of the two functions is still very small, the connection strengths calculated this way are probably still very much underestimated.

So far, only connections between stimulated neurons and spike units on electrodes have been considered. To estimate functional connectivity maps in a similar way to the baseline connectivity maps, these connections have to be converted to electrode–electrode connections. To this end, all valid connections produced by one stimulation episode are considered. If there are valid connections from the stimulation site to a spike unit on more than one electrode, the connection strength between these electrodes is increased by one. Finally, connection strengths are normalized by the duration of the holographic stimulation experiment under the assumption that longer experiments with more stimulated neurons will, on average, have higher resulting connection strength values in a well-connected network. For comparison between days, all connection strengths were also normalized to the maximum value across days.

One main result of the holographic single-neuron stimulation is the set of all valid connections $n_{ijkk}$ of the neuron stimulated in episode $i$ to the unit $j_k$ on recording electrode $k$. The number of functional connections per neuron is then:

$$N_{conn,neuron,i} = \sum_{j_k,k} n_{ij_k k}.$$

The number of valid connections per electrode, on the other hand, is then:

$$N_{conn,electrode,k} = \sum_{i,j_k} n_{ij_k k}.$$

## Data Availability

All data are available from the corresponding authors upon reasonable request.

## Supplementary Information

## Acknowledgements

The authors thank Dr. Karl Farrow for critical feedback on the manuscript. J Striebel acknowledges support by the Joachim Herz Foundation. J Czarske was supported by the Deutsche Forschungsgemeinschaft (DFG CZ 55/39-1 and the Reinhart Koselleck Project for High-Risk Research). V Busskamp acknowledges funding from the Volkswagen Foundation (Freigeist–A110720), the European Research Council (ERC-StG-678071–ProNeurons), the Deutsche Forschungsgemeinschaft (BU 2974/4-1, SPP2127 and EXC-2151-390873048 – Cluster of Excellence–ImmunoSensation[2] at the University of Bonn), the Pro Retina Foundation, and the Paul Ehrlich Foundation.

### Author Contributions

F Schmieder: data curation, software, formal analysis, validation, investigation, visualization, methodology, and writing—original draft, review, and editing.
R Habibey: resources, data curation, software, formal analysis, validation, investigation, visualization, methodology, and writing—original draft, review, and editing.
J Striebel: validation, and writing—original draft, review, and editing.
L Buttner: conceptualization, supervision, funding acquisition, methodology, project administration, and writing—review and editing.
J Czarske: conceptualization, resources, supervision, funding acquisition, and writing—review and editing.
V Busskamp: conceptualization, resources, supervision, funding acquisition, project administration, and writing—review and editing.

### Conflict of Interest Statement

The authors declare that they have no conflict of interest.

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
