## [Reviewer comments · Life Science Alliance]

Life Science Alliance

Tracking connectivity maps in human stem-cell-derived neuronal networks by holographic-optogenetics

Felix Schmieder, Rouhollah Habibey, Johannes Striebel, Lars Büttner, Jürgen Czarske, and Volker Buskamp
DOI: <https://doi.org/10.26508/lsa.202101268>

Corresponding author(s): Volker Buskamp, Universitäts-Augenklinik Bonn and Jürgen Czarske, TU Dresden

Review Timeline:

Submission Date:	2021-10-21
Editorial Decision:	2021-11-25
Revision Received:	2022-02-22
Editorial Decision:	2022-03-23
Revision Received:	2022-03-25
Accepted:	2022-03-29

Scientific Editor: Novella Guidi

Transaction Report:

November 25, 2021

Re: Life Science Alliance manuscript #LSA-2021-01268

Prof. Volker Buskamp
Universitäts-Augenklinik Bonn
Dep. of Ophthalmology
Ernst-Abbe-Straße 2
Bonn 53127
Germany

Dear Dr. Buskamp,

Thank you for submitting your manuscript entitled "Tracking connectivity maps in human stem-cell-derived neuronal networks by holographic-optogenetics" to Life Science Alliance. The manuscript was assessed by expert reviewers, whose comments are appended to this letter. As you will note from the reviewers' comments below, all the reviewers are somewhat positive and excited about the work. However, they do raise some concerns that would need to be addressed in the revised version before resubmission. Please address the common concerns of rev#1 and rev#2 regarding the lack of single-cell spike sorting analysis, spike rasters of population and clarification of network analysis, as in its current form it is unclear how you have approached the analysis. All the other concerns raised by the reviewers should be addressed as well. We, thus, encourage you to submit a revised version of the manuscript back to LSA that responds to all of the reviewers' points.

Thank you for this interesting contribution to Life Science Alliance. We are looking forward to receiving your revised manuscript.

Sincerely,

B. MANUSCRIPT ORGANIZATION AND FORMATTING:

Reviewer #1 (Comments to the Authors (Required)):

Schmieder et al employed a highly precise holographic optical stimulation to target and activate individual neurons expressing Chr2-EYFP to reveal functional neural network connectivity maps. Single-neuron stimulation was contrasted with full-field optical stimulation - a more basic application of optogenetics in which all neurons are simultaneously stimulated. The hiPSC 'iNGN' derived neurons were matured for up to about ~80 days for optogenetic manipulations, and electrical activity was recorded regularly using multi-electrode arrays.

Using holographic optogenetic stimulation, the authors extracted temporal and spatial data regarding the propagation of activity across neuronal networks. This is potentially very useful.

Unless I misunderstood, the authors performed the analysis of the entire paper on recordings from four MEA wells. This seems to be a substantial weakness to appreciate the method's reliability and fully capture the variance of functional connectivity maps. Another technical weakness, which I believe would significantly improve the paper if addressed, is the lack of single-cell spike sorting analysis throughout most of the paper.

The authors show that holographic optical stimulation of individual neurons can be used to elucidate neuronal network connectivity - it's very cool! But, unfortunately, after reading this draft of the paper, I was left unsure whether or not it is worthwhile the extra technical effort to set up holographic optical stimulation rather than full-field optogenetic stimulation or even using electrode stimulation.

Please see below more specific comments and suggestions by section, which hopefully will help strengthen the study:

Abstract.

The abstract could be a bit more precise. For example:

1- P2, line 37: 'By optogenetic stimulation, we detected an earlier onset of neuronal responses'. Earlier than what? What are 'neuronal responses'?

2- P2, line 40-43. What functional connectivity motifs and long-term dynamics were revealed? What value do these motifs and dynamics have for 'establishing hiPSC-derived neurons as profound functional testbeds for basic and biomedical research'?

Introduction

3- P4, line 92-93: The rationale that most hiPSC-derived neuron studies provide a "snapshot information on action potential frequencies", and that instead "in-depth information is required" is a bit vague.

Results

4- Many electrodes are inactive in this data-set, which seems to lower the average values artificially. For example, less than 1Hz AP frequency for the control is relatively low.

In figure 2C the authors show spike sorted traces as examples. However, all the properties in Fig 2D-G should be plotted per active neurons rather than per electrode. The authors should probably also show the percentage of active electrodes in the main figure and ideally the number of active neurons per MEA well with and without stimulation.

5- Fig 2D and P6, line 162: '... a peak in activity followed by a decline...' It is not clear if the sudden drop in activity at 70dpi is biological as suggested or simply an experimental bias. Would this decrease in activity be observed in multiple plates, each cultured independently? These data should be replicated at the very least twice in independent experiments (different batches of cells).

6- Figure 2D, E, G. I find the break in the y-axis a bit confusing.

7- The authors might want to consider including representative raster and spike plots displaying MEA activity across dpi's with/without stimulation and show the action potential firing (or burst) synchronised with light flashes. It may be slightly confusing that the light stimulation was used at 0.5 Hz, yet the reported firing is 2.69Hz. Spike sorted analysis would also help here.

8- Can you specify in fig 1 and results what promoter was used to express ChR2-EYFP?

9- P6, line 168: Change 'wavefronts' to 'waveforms'.

10- P7, line 205: '...from local and sparse APs to synchronised burst activities over time... The authors should include a measure of synchronicity at each dpi in figure 2. I would also suggest that the authors include measurements of synchronised burst events (A.K.A. network events, network bursts, population spiking, or population events) frequency and duration for the MEA wells to demonstrate the formation of mature neuronal networks in the cultures.

11 - On the electrodes shown in Fig S4, there seem to be two spikes triggered by the 50ms light pulse. Is 50ms too long?

12- P7, lines 206-208: The authors state that: 'Optogenetic stimulation ... led to increased firing rates at earlier developmental timepoints, suggesting that optogenetic stimulation boosts neuronal activity of hiPSC-derived neurons'. I think this conclusion is somewhat misleading. It might sound like the application of optogenetic stimulation boosted the maturation of baseline neuronal activity. Otherwise, although important to demonstrate as a proof of concept, it is expected that optogenetic stimulation should increase firing rates during stimulation. Instead, the data show that some neurons present in cultures are capable of firing action potential but are spontaneously silent, at least during the time frame of the recording.

13- Figure 3D. Is the AP frequency of each episode the average frequency across all responding electrodes? What is the dpi for these recordings?

14- Fig 3C. Given the variance in the percentage of active electrodes, four replicate wells seem insufficient to perform conclusive statistics for any analysis per well.

15- P8, line 230: '... weaker network activity...' The wording should probably be changed. The network activity is not necessarily "weaker". Instead, holographic single-cell stimulation results in the activation of smaller sub-networks within the neuronal population than full-field optogenetic stimulation activating all optogene expressing neurons and their networks across the entire population.

16- P8 line 232: I don't understand what the authors mean by "...but precise AP frequency trends were revealed in our data".

17- Are Fig 3 E, F, G showing the same data but plotted differently?

18 - Again, single-cell data should be presented in terms of AP frequencies

19 - Fig 3. P-value should probably be shown even if not significant.

20- The tracking of functional connectivity maps is very interesting for its potential application in disease models. I also understand the potential of holographic optogenetics for getting connectivity information from neurons too far away from stimulating or recording electrodes. However, considering the extra technical burden of setting up holographic stimulation, it is unclear if the extra information obtained justifies the extra work. This is not apparent from the analysis performed by the author, in my opinion. In addition, the authors suggest that there is a solid dynamic component of the map over time in vitro. The extent of variance between maps between replicates should be clearly shown with multiple experiments and replicate MEA wells. If too variable and too dynamic, this could be a significant shortcoming of applying this analysis to disease models.

Discussion

21- P14, line 383 - the authors claim the astrocytes on coverslips provided a stable microenvironment, but they have not compared their astrocyte-coverslip system with standard culture methods in this paper. So probably not relevant for discussion.

Conclusion

22- Line 367-369 This is a bit of an overstatement... or misleading at best. This is not the only paper looking at hiPSC-derived neurons over 70 days in culture and other functional properties than neuronal firing rates.

23- P16, line 459 - the authors claim the holographic optogenetics system can be used to control different neurons with various wavelengths but presented no experimental evidence for this.

Methods

24- A few publications have described phototoxicity of tissue culture media, including on hiPSC-derived neurons (e.g., PMID: 33144563). The authors should discuss this as a potential limitation of their current experimental design.

Supplementary

25- Scale bars for images are missing

26- P22, line 678 refers to S11, but the supplied supplementary figures only go up to 9?

Reviewer #2 (Comments to the Authors (Required)):

The manuscript by Schmieder et al titled "Tracking connectivity maps in human stem-cell-derived neuronal networks by holographic-optogenetics" aims to study the development of functional connections in human iPSC-derived neuronal cultures using spatially patterned optogenetic stimulation and MEA recordings. The paper develops on existing methods of neuron-glia cultures incorporating spatially patterned optogenetics and repeated recordings of human neuronal cultures over 80 days in vitro.

This manuscript is in the scope of Life Science Alliance, which includes methods and resources, as well as descriptive data. There is experimental evidence supporting the methods of optogenetic stimulation, but a number of details that are left out make assessing other conclusions of the paper challenging. The critical issues, which are mostly related to analysis are detailed below.

Key issues to address:

1. Details on spike sorting, which is essential for multi-electrode analysis, particularly in the area of network connectivity is absent in the manuscript. The mention of this is in line 198 "The AP waveforms of several neurons were extracted by spike sorting" and line 167. In the supplemental figures (Fig. S3) it seems that amplitude was used to identify units?. But based on the distribution of amplitudes of neuron 2 and 3 in the holographic stimulation, a number of spikes in neuron 3 could be assigned to neuron 2. What was used to validate this criterion? Why did the authors not use traditional mixture of gaussians models (Lewicki 1998), or more recent template matching approaches (Pachitariu et al NeuRIPS 2016, Yger et al 2018, but also see others) that would produce potentially better sorting. What criterion was used to estimate unit isolation (waveform shape, interspike interval distribution, etc) once spikes had been sorted?
2. Example spike rasters of population activity would be very useful to visualize the temporal structure of activity across individual neurons, and also the extent to which this is correlated across time. Additionally, a summary of the distribution of units in each channel, and the change in this distribution over time would be critical.
3. Clarification of network analysis due to issue 1. Based the spike sorting that has been described, this complicates the network connectivity analysis because changes in firing rate and connectivity are intermixed in their analysis. The raw traces in the channel from experiment e1 in figure 3 appear to have multiple units (look to be different waveforms). Thus, an increase in AP could be because a channel records more neurons (number of units in a channel) or a single neuron is firing more, differences it is unclear if their analysis can clarify. Additionally, single neurons can often be detected on multiple channels in MEA cultures (Chichlinsky lab, and others), and as such, functional synchrony could be a detection artifact. These issues are not clarified.
4. The burst analysis is somewhat confusing in part because of the issues about. How much of this effect can be attributed to recording from additional neurons as they mature and their responses are detected by the electrode. These details should be clarified.
5. The connectivity graphs are compelling (Fig. 6 B and C) as they appear to recapitulate work by Cossart et al on the relationship between spontaneous patterns and stimulated activity. The major claim of the authors is that the connectivity changes with development age, as illustrated in Fig. 6B. Could this be due just to spurious correlations from changes in firing rate (see Doiron et al Nature). Shuffling the spike times and then calculating the connectedness would go a long way in addressing this concern.

Reviewer #3 (Comments to the Authors (Required)):

Combining optogenetics with neural networks in iPSC-derived organoids is a promising approach to tune and interrogate such model circuits. One common caveat of such networks is immaturity and electrogenic activity that limit their resemblance of mature physiological networks. Therefore, efforts to generate more physiological network behavior are of great value. While recently there has been some progress, this tour de force multidisciplinary study advances the field by moving toward more selective and versatile holographic optogenetic stimulation within the organoids. While holographic optogenetic stimulation has been applied in

slices and intact brain, I think the scope of this study is novel and offers an avenue for how to arrive at more physiological neural networks in organoids. Given the potential of organoids for disease modeling and drug screening this advance has major relevance.

The study is quite impressive and comprehensive, the methods sound and the data of high quality. The MS has been carefully prepared. I mostly have minor edits which might help to further improve this MS.

I think the holographic stimulation work of Valentina Emiliani and Shy Shoham deserve better representation in intro/discussion. Also I found the statement starting from line 104 unclear. Moreover, the identification of the individual neurons in the present study in comparison with 2plsm imaging studies could be introduced already in the abstract and/or last section of introduction.

"We added the excitatory glutamatergic synaptic blockers NBQX" consider rephrasing, e.g. glutamate receptor blocker...

Line 191: "poly-synaptic functional neuronal networks" there is evidence for synaptic transmission but the authors have no control on the question of across how many synapses the network is built: "synaptically connected functional neuronal networks" would probably be more appropriate

"Combined holographic stimulation with MEA recordings 212 provided high spatial stimulation resolution of 8 μm spots [52]."

Would be great to show some evidence for the spatial resolution here or in supp.

Fig. 3: consider using blue color instead of amber for the full field, to not confuse the reader.

227: "moderate- and high-frequency APs." This should be defined

231: "but precise AP frequency trends were revealed in our data" this is pretty useless without specifying what you are talking about

260: "A shorter delay was also attributed to a direct neuronal activation, as shown by a correlation analysis of high PSTH peak amplitudes with short latencies (Figure 4E)." this notion would be best asserted by recordings with NBQX+APV, the statement of line 264 should only be made if documented by this experiment

Section 2.5 needs some work: if I get it right the authors perform in-depth analysis of a single recording with exemplary responses to then arrive at the conclusion that holographic optogenetic stimulation but not full-field optogenetic stimulation revealed a specific motif.

While I understand that the authors are excited here, I do find this a bit weak as evidence supporting their conclusion. The authors need to provide additional data to support the notion of direct and indirect stimulation and might want to use NBQX+APV exposure as further test.

Re: Life Science Alliance manuscript #LSA-2021-01268

Thank you for submitting your manuscript entitled "Tracking connectivity maps in human stem-cell-derived neuronal networks by holographic-optogenetics" to Life Science Alliance. The manuscript was assessed by expert reviewers, whose comments are appended to this letter. As you will note from the reviewers' comments below, all the reviewers are somewhat positive and excited about the work. However, they do raise some concerns that would need to be addressed in the revised version before resubmission. Please address the common concerns of rev#1 and rev#2 regarding the lack of single-cell spike sorting analysis, spike rasters of population and clarification of network analysis, as in its current form it is unclear how you have approached the analysis. All the other concerns raised by the reviewers should be addressed as well. We, thus, encourage you to submit a revised version of the manuscript back to LSA that responds to all of the reviewers' points.

We would like to thank the Reviewers for their insightful comments to improve the quality of our manuscript. In the revised manuscript, we have addressed all points.

Briefly, we have performed single-cell spike sorting analyses for spontaneous activity, full-field stimulation, holographic optogenetic stimulation, and the synaptic inhibition experiments. We have updated all figures and text accordingly. We now show neuron-based data (new analysis, main figures) and electrode-based data (initial version of the manuscript, supplementary file). Raster plots of neuronal activity are included both for neuronal data and electrode data. The effects on active neuron numbers and their firing rates by full-field and holographic stimulation were analyzed and included. We have updated and revised the text, figures and the supplementary file according to the Reviewers comments and questions and thereby improved clarity. We have put an emphasis on explaining the exemplar connectivity motif extraction.

Please find all of our responses in blue font color interspersed with the Reviewer comments.

- You will be guided to complete the submission of your revised manuscript and to fill in all necessary information. Please get in touch in case you do not know or remember your login name.
 - While you are revising your manuscript, please also attend to the below editorial points to help expedite the publication of your manuscript. Please direct any editorial questions to the journal office.
 - The typical timeframe for revisions is three months. Please note that papers are generally considered through only one revision cycle, so strong support from the referees on the revised version is needed for acceptance.
 - When submitting the revision, please include a letter addressing the reviewers' comments point by point.
 - We hope that the comments below will prove constructive as your work progresses.
-

- A letter addressing the reviewers' comments point by point.
- An editable version of the final text (.DOC or .DOCX) is needed for copyediting (no PDFs).
- High-resolution figure, supplementary figure and video files uploaded as individual files: See our detailed guidelines for preparing your production-ready images, <https://www.life-science-alliance.org/authors>
- Summary blurb (enter in submission system): A short text summarizing in a single sentence the study (max. 200 characters including spaces). This text is used in conjunction with the titles of papers, hence should be informative and complementary to the title and running title. It should describe the context and significance of the findings for a general readership; it should be written in the present tense and refer to the work in the third person. Author names should not be mentioned.

B. MANUSCRIPT ORGANIZATION AND FORMATTING:

Reviewer #1 (Comments to the Authors (Required)):

Schmieder et al employed a highly precise holographic optical stimulation to target and activate individual neurons expressing Chr2-EYFP to reveal functional neural network connectivity maps. Single-neuron stimulation was contrasted with full-field optical stimulation - a more basic application of optogenetics in which all neurons are simultaneously stimulated. The hiPSC 'iNGN' derived neurons were matured for up to about ~80 days for optogenetic manipulations, and electrical activity was recorded regularly using multi-electrode arrays.

Using holographic optogenetic stimulation, the authors extracted temporal and spatial data regarding the propagation of activity across neuronal networks. This is potentially very useful.

We would like to thank the Reviewer for acknowledging our work and their detailed constructive review.

Unless I misunderstood, the authors performed the analysis of the entire paper on recordings from four MEA wells. This seems to be a substantial weakness to appreciate the method's

reliability and fully capture the variance of functional connectivity maps. Another technical weakness, which I believe would significantly improve the paper if addressed, is the lack of single-cell spike sorting analysis throughout most of the paper.

We appreciate the Reviewer's comments and agree that N= 4 MEAs is not extremely high but still, it provides sufficient statistical power for our analyses. Please keep in mind that our data was gained over an extensive long-term experimental process. In addition, we collected our data based on repeated experiments and recordings at five different time points (dpi 35, 45, 60, 70 and 80). Therefore, comparisons were performed at 5 time points that enhanced the statistical power of the obtained results.

In our initial manuscript, the recorded data were partially sorted to extract neurons (mainly for holographic data in Figures 5, 6 and S6). In the revised manuscript, we sorted spikes for all conditions (spontaneous activity and full-field stimulation in Figure 2 and S2, holographic stimulation in Figure 3, as well as recordings of chemical treatment experiments Figure S4 and S5). All data are presented both in the electrode-based manner (Fig S2, S4) and in sorted spikes (Fig 2, 3, S5).

The authors show that holographic optical stimulation of individual neurons can be used to elucidate neuronal network connectivity - it's very cool! But, unfortunately, after reading this draft of the paper, I was left unsure whether or not it is worthwhile the extra technical effort to set up holographic optical stimulation rather than full-field optogenetic stimulation or even using electrode stimulation.

We thank the reviewer for their excitement for holographic stimulation. We are convinced that these kinds of experiments provide deeper insights into neuronal circuits studied with MEA technology. We want to point out that the selectivity of holographic optogenetic stimulation on targeting individual neurons in the network is not possible with conventional electrical stimulation using the recording electrodes. We have improved the clarity on this aspect in the manuscript (introduction second paragraph and discussion 5th paragraph). Based on this work, we aim at designing complex experiments with a combination of optogenetic actuators (inhibitory and excitatory) and neuronal cell types (inhibitory and excitatory). We agree that this comes with additional technical demands, but since more and more commercially available holographic stimulation systems become available, this obstacle will likely be substantially reduced.

Please see below more specific comments and suggestions by section, which hopefully will help strengthen the study:

Abstract.

The abstract could be a bit more precise. For example:

1- P2, line 37: 'By optogenetic stimulation, we detected an earlier onset of neuronal responses'. Earlier than what? What are 'neuronal responses'?

We agree and have updated the entire abstract including this wording for clarity.

2- P2, line 40-43. What functional connectivity motifs and long-term dynamics were revealed? What value do these motifs and dynamics have for 'establishing hiPSC-derived neurons as profound functional testbeds for basic and biomedical research'?

Changes were applied to this part and the data regarding the number and strength of functional connections was added. The wording for the last sentence was also changed to fit the scope of the manuscript: "Single-cell holographic stimulation facilitated to trace propagating evoked activities of 400 individually stimulated neurons per MEA. Thereby, we revealed precise functional connectivity motifs between neurons. Holographic stimulation data on different days demonstrated an increase in the number and strength of connections with culture age. "

Introduction

3- P4, line 92-93: The rationale that most hiPSC-derived neuron studies provide a "snapshot information on action potential frequencies", and that instead "in-depth information is required" is a bit vague.

We have updated this part and now provide more information. "However, in-depth analyses of long-term network functional connectivity features were not revealed as electrical stimulation of individual neurons through MEA electrodes is challenging. Stimulation artifacts impede the extraction of data during and after the applied electrical pulses through the recording electrodes [28]. Low-density neuronal cultures on high-density MEAs enabled to electrically trigger individual neuronal activity [29], however electrical stimulation of single neurons in densely connected networks is often challenging. High-density MEAs substrates are not transparent that limit the live imaging morphological details of the network at single neurons resolution. In addition, neuronal activity cannot be inhibited by MEA electrodes [28]." In this work, we provide a methodological approach to studying functional data with higher resolution. Compared to full-field stimulation, our holographic stimulation data facilitated tracking functional connectivity motifs and connectivity maps of individual cultures over time. Such continuous recordings combined with holographic stimulation on *in vitro* hiPSCs-networks have not been shown before. Parallel evaluations of neuronal firing rates and burst activities (as has been done in previous works) with network functional connectivity data (i.e. strength of connections, number of connections, structure-function relationship, ...) is key to dissect and to study in-depth diseased and healthy neuronal networks. Our study represents a technical proof-of-concept that all these kind of data can be collected at once. In addition, a higher number of active neurons including a faster functional network maturation can be achieved by optogenetic activation.

Results

4- Many electrodes are inactive in this data-set, which seems to lower the average values artificially. For example, less than 1Hz AP frequency for the control is relatively low.

For the AP frequency analysis, we have excluded the electrodes that did not show any activity for the whole course of the experiment (80 dpi). However, if an electrode showed activity in some time points and was inactive in others, we had to attribute 0 value for non-active points. We applied the same procedure for comparing stimulated and unstimulated conditions. This was important to include the electrodes with lower activity to have a correct estimation of developing functional features with culture age. Also, new parts regarding neuronal firing rate and

distribution of neurons firing at different frequencies have been added to Figure 2 and text (results section 2.2).

In figure 2C the authors show spike sorted traces as examples. However, all the properties in Fig 2D-G should be plotted per active neurons rather than per electrode. The authors should probably also show the percentage of active electrodes in the main figure and ideally the number of active neurons per MEA well with and without stimulation.

We would like to thank the Reviewer for this feedback. To this end, the corresponding analysis for Figure 2 was repeated using sorted signals and the neuronal units. AP frequency and burst frequency in each neuron were calculated and applied in this analysis. The distribution of neurons firing at different frequencies was also plotted. The number of detected neurons (active neurons) was calculated and compared between spontaneous activity and optogenetically-stimulated conditions. The electrode-based activity results were shifted to the supplementary Figure S2. A raster plot of recorded APs in neurons is now added to Figure 2.

5- Fig 2D and P6, line 162: '... a peak in activity followed by a decline...' It is not clear if the sudden drop in activity at 70dpi is biological as suggested or simply an experimental bias. Would this decrease in activity be observed in multiple plates, each cultured independently? These data should be replicated at the very least twice in independent experiments (different batches of cells).

We have observed this activity profile in all MEAs both on percentage of active electrodes and as well on the number of detected neurons (as it has been mentioned in Fig 3 E and G). In addition, other independent experiments on iNGN neurons in our group have shown a similar profile with a peak around dpi 60 followed by decline (unpublished data, see below).

6- Figure 2D, E, G. I find the break in the y-axis a bit confusing.

We thank the Reviewer for pointing this out. The range of data distribution is very large. Therefore, bringing data in a single Y-axis without a break leads to a lack of visibility for major parts of data. In the new version we adjusted the y-axis based on the logarithmic scale (Log 10) therefore all data are visible.

7- The authors might want to consider including representative raster and spike plots displaying MEA activity across dpi's with/without stimulation and show the action potential firing (or burst) synchronised with light flashes. It may be slightly confusing that the light stimulation was used at 0.5 Hz, yet the reported firing is 2.69Hz. Spike sorted analysis would also help here.

It is true that the light stimulation frequency was lower than 2.69 Hz, however this is not distributed equally for the whole length of the recording period. Therefore, the optically evoked activity is simply detectable (for example in raster plot or by PSTH analysis). To make this point clearer, we added raster plots of activity for spontaneous and full-field stimulation activity of neurons (Figure 2) and electrodes (Figure S2), as well as holographic stimulation of neurons (Figure 3). The same was done for activity profiles under NBQX-APV treatment with representative raster plots for all conditions (baseline, treatment, and washout; Figure S5). Coupled electrode and neuronal responses to applied pulses of full-field stimulation are represented in these figures.

8- Can you specify in fig 1 and results what promoter was used to express Chr2-EYFP?

We have used the elongation factor 1 α (ef1 α) promotor element to obtain a strong Chr2-EYFP expression. We added this information to the figure and figure legend (Figure 1).

9- P6, line 168: Change 'wavefronts' to 'waveforms'.

Many thanks, we have corrected this.

10- P7, line 205: '...from local and sparse APs to synchronised burst activities over time... The authors should include a measure of synchronicity at each dpi in figure 2. I would also suggest that the authors include measurements of synchronised burst events (A.K.A. network events, network bursts, population spiking, or population events) frequency and duration for the MEA wells to demonstrate the formation of mature neuronal networks in the cultures.

As suggested by the Reviewer, we included raster plot data from spontaneous activity in electrodes and as well in the sorted neuronal units shows episodes of synchronous firing across the network (Figure 2, Figure S2 and Figure S5). In addition, chemical treatment with NBQX-APV significantly reduced the network activity (no burst activity was observed in presence of blockers), which was related to the blockage of the synaptic communication. The development of burst activity features shown in synchronized raster plots (Figure 2) and results of chemical treatment with NBQX-APV (Figure S4 and Figure S5) altogether confirm the synaptic communication and network functional maturation. The text in result section 2.2 and as well in discussion has been updated.

11 - On the electrodes shown in Fig S4, there seem to be two spikes triggered by the 50ms light pulse. Is 50ms too long?

Normally, the kinetics of the ChR2 channels are very fast and one neuron may respond with more spikes during the 50ms period. As we have shown in our previous work, full-field stimulation of hiPSC-derived networks with 1 ms pulses and 25 Hz triggered neuronal AP frequencies around 25Hz (Klapper, S. D. *et al. Sci. Rep.* **7**, 1–9, 2017). However, here applying shorter pulses with holographic stimulation did not induce responses in most cases. Therefore, we selected longer pulses, accepting that in some cases we observe two responses in one neuron during the 50ms.

12- P7, lines 206-208: The authors state that: 'Optogenetic stimulation ... led to increased firing rates at earlier developmental timepoints, suggesting that optogenetic stimulation boosts neuronal activity of hiPSC-derived neurons'. I think this conclusion is somewhat misleading. It might sound like the application of optogenetic stimulation boosted the maturation of baseline neuronal activity. Otherwise, although important to demonstrate as a proof of concept, it is expected that optogenetic stimulation should increase firing rates during stimulation. Instead, the data show that some neurons present in cultures are capable of firing action potential but are spontaneously silent, at least during the time frame of the recording.

We appreciate the Reviewer's input regarding this point. We have updated the text based on the comments and included new data to support this. We concluded the beneficial effects of full-field optogenetic stimulation on improving network functional maturation based on its effect on engaging all activated neurons simultaneously. We show that during full-field stimulation more neurons show activity compared to spontaneous activity (Figure 2E). Based on the sorted data, we observed that the number of sorted neurons significantly increases by full-field optogenetic stimulation. This engages more neurons in the network activity. In addition, our data showed that the firing rate of individual neurons was significantly increased by full-field optogenetic stimulation. Therefore, Optogenetic stimulation both engages spontaneously silent neurons (Figure 2E) and increases the firing frequency of the activated neurons (Figure 2F). Raster plots of activity showed that full-field stimulation activated all neurons simultaneously which is critical for network maturation.

13- Figure 3D. Is the AP frequency of each episode the average frequency across all responding electrodes? What is the dpi for these recordings?

This plot is only for one electrode to represent how its activity is affected only in some of the episodes (blue squares). We separated the responses to active episodes (episodes that a neuron responded with significantly higher frequency, please check section 3.7.3 and 3.7.9 in the Methods) and compared them with all other episodes (Figure 3H). The average AP frequency for all electrodes has been plotted in Supplementary Figure S6. This experiment has been performed at 35 dpi. This information was added to the figure legend.

14- Fig 3C. Given the variance in the percentage of active electrodes, four replicate wells seem insufficient to perform conclusive statistics for any analysis per well.

Based on a recent review by Negri et al (2020 eNeuro), naturally MEA electrophysiology data show higher variance between wells and between electrodes of the same MEA. Therefore, the variance in the percentage of active electrodes and the firing frequency is inevitable. Regarding the long-term experiment and long-duration of the holographic stimulation experiments, we performed the analysis in four cultures. To compare spontaneous activity vs. full-field stimulation and holographic stimulation vs. full-field stimulation data we used electrode-based or neuron-based data with larger sample sizes ($n > 90$). Holographic stimulation was performed at single-neuron levels and data was collected from individual episodes therefore statistical analysis was performed in statistically sufficient numbers of neurons ($n = 239$) and episodes (more than 400 episodes per culture per day). For the effect of distance on PSTH profile we pooled data from five sessions of holographic stimulation per culture (35 to 80 dpi) each session including more than 400 episodes. Therefore, we are convinced that regardless of the number of MEA wells, the higher number of the electrodes, neurons and episodes per day and repeating the experiment per MEA at 5 different time points gave enough power to our statistical evaluations.

15- P8, line 230: '... weaker network activity...' The wording should probably be changed. The network activity is not necessarily "weaker". Instead, holographic single-cell stimulation results in the activation of smaller sub-networks within the neuronal population than full-field optogenetic stimulation activating all optogene expressing neurons and their networks across the entire population.

We agree with the Reviewer and have changed the wording to: "In general, holographic stimulation resulted in more local activity compared to full-field stimulation which likely results in the activation of all optogene-expressing neurons and their respective networks across the sample,..."

16- P8 line 232: I don't understand what the authors mean by "...but precise AP frequency trends were revealed in our data".

This was related to Figure 3 E (initial version). We deleted this part. New data was added to Figure 3 including Figure 3G-I. We also changed the corresponding information in the text based on new data (please see section 2.3 of results).

17- Are Fig 3 E, F, G showing the same data but plotted differently?

These parts of Figure 3 were deleted, and new data were added including neuronal unit data (Figure 3 G-I).

18 - Again, single-cell data should be presented in terms of AP frequencies

We sorted the signals and extracted neuronal units (details of sorting method is in section 3.7.8) then the AP frequency was calculated in each sorted neuron separately per day (Figure 3H). The average of AP frequency in the sorted neurons of an electrode was also included in Figure 3F to compare between full-field and holographic stimulation responses.

19 - Fig 3. P-value should probably be shown even if not significant.

P-values for the non-significant comparisons were included in each graph (Figure 3E and G).

20- The tracking of functional connectivity maps is very interesting for its potential application in disease models. I also understand the potential of holographic optogenetics for getting connectivity information from neurons too far away from stimulating or recording electrodes. However, considering the extra technical burden of setting up holographic stimulation, it is unclear if the extra information obtained justifies the extra work. This is not apparent from the analysis performed by the author, in my opinion. In addition, the authors suggest that there is a solid dynamic component of the map over time *in vitro*. The extent of variance between maps between replicates should be clearly shown with multiple experiments and replicate MEA wells. If too variable and too dynamic, this could be a significant shortcoming of applying this analysis to disease models.

Advantages of holographic optogenetic stimulation on extracting detailed network functional data at single-cell resolution have already been exploited on *in vivo* and brain slices, but have not been tried in the *in vitro* models. Here we used our custom-made setup, however commercial holographic stimulation systems are emerging which will be more straightforward to setup and use in different *in vitro* applications. The main goal of the current method work is to adapt holographic stimulation to the functional evaluation of hiPSCs-derived networks at the single neuron level and extract fine connectivity maps. This approach can be strengthened by integrating it with high-density MEAs and multiple optogenetic actuators to unravel complex functional properties of *in vitro* circuits and use these data for tracking the functional phenotype of neural circuits. *In vivo* network functional connectivity maps are highly dynamic and affected by a multitude of parameters including sensory information and structural changes. Developing *in vitro* networks undergo both functional and structural changes over time which also affect its functional connectivity maps over time as we observed in our data. Overall, the variance of the functional connectivity data was high between the number of connections per electrode (Figure 6E), while connections per neuron and connection strength showed a lower extent of variance (Figure 6F and 6G). This could be related to the different number of detected neurons per electrode (Figure 2E and 3G), that comes from a randomly distributed network structure. Engineered *in vitro* circuits with predefined numbers and positions of neurons and their connections could be integrated with our holographic platform to improve the quality of functional connectivity data.

Discussion

21- P14, line 383 - the authors claim the astrocytes on coverslips provided a stable microenvironment, but they have not compared their astrocyte-coverslip system with standard culture methods in this paper. So probably not relevant for discussion.

The discussion regarding the effect of astrocytes on neuronal activity has been modified in the text.

Conclusion

22- Line 367-369 This is a bit of an overstatement... or misleading at best. This is not the only paper looking at hiPSC-derived neurons over 70 days in culture and other functional properties than neuronal firing rates.

We agree with the Reviewer and have updated these lines in the first paragraph of the discussion.

23- P16, line 459 - the authors claim the holographic optogenetics system can be used to control different neurons with various wavelengths but presented no experimental evidence for this.

We thank the Reviewer for pointing out this misleading statement. This was meant as an outlook at the end of the discussion. As we stated that “In addition to the stimulation of excitatory neurons, one can also use holographic illumination to exploit inhibitory optogenetic actuators for silencing neurons.”

Methods

24- A few publications have described phototoxicity of tissue culture media, including on hiPSC-derived neurons (e.g., PMID: 33144563). The authors should discuss this as a potential limitation of their current experimental design.

We added a section discussing the topic at the end of the discussion.

Supplementary

25- Scale bars for images are missing

We thank the Reviewer for pointing this out. The distance between electrodes is 200 μm . We added the information to the Figure legends in the main text and supplementary file.

26- P22, line 678 refers to S11, but the supplied supplementary figures only go up to 9?.

We appreciate the Reviewer for pointing this out. In the initial version of the manuscript, the supplementary section 11 (S11) was missing from the supplement files. The numbering was corrected. Furthermore, in the current version of the manuscript supplementary section 12 (S12) was added describing the method and formulas to calculate the connectivity maps based on spontaneous activity.

Reviewer #2 (Comments to the Authors (Required)):

The manuscript by Schmieder et al titled "Tracking connectivity maps in human stem-cell-derived neuronal networks by holographic-optogenetics" aims to study the development of functional connections in human iPSC-derived neuronal cultures using spatially patterned optogenetic stimulation and MEA recordings. The paper develops on existing methods of neuron-glia cultures

incorporating spatially patterned optogenetics and repeated recordings of human neuronal cultures over 80 days in vitro.

This manuscript is in the scope of Life Science Alliance, which includes methods and resources, as well as descriptive data. There is experimental evidence supporting the methods of optogenetic stimulation, but a number of details that are left out make assessing other conclusions of the paper challenging. The critical issues, which are mostly related to analysis are detailed below.

We would like to thank the Reviewer for acknowledging our work and for providing constructive feedback that we have included in our revised manuscript.

Key issues to address:

1. Details on spike sorting, which is essential for multi-electrode analysis, particularly in the area of network connectivity is absent in the manuscript. The mention of this is in line 198 "The AP waveforms of several neurons were extracted by spike sorting" and line 167. In the supplemental figures (Fig. S3) it seems that amplitude was used to identify units?. But based on the distribution of amplitudes of neuron 2 and 3 in the holographic stimulation, a number of spikes in neuron 3 could be assigned to neuron 2. What was used to validate this criterion? Why did the authors not use traditional mixture of gaussians models (Lewicki 1998), or more recent template matching approaches (Pachitariu et al NeuRIPS 2016, Yger et al 2018, but also see others) that would produce potentially better sorting. What criterion was used to estimate unit isolation (waveform shape, interspike interval distribution, etc) once spikes had been sorted?

We would like to thank the Reviewer for pointing towards spike sorting. We have added further details on the used spike sorting algorithm in the methods section 3.7.8. As described by Chaure et al. (2018), spike sorting with wave_clus3 is based on a wavelet transformation of order 4 of the spike waveforms. Wavelet coefficients are used as parameters for sorting instead of the often-used principal components. As far as we understood, the amplitude is important for the clustering but by no means an exclusive feature, as is the case for the PCA. All spikes in a cluster have an inter-spike interval of at least 3ms. At one point, we had to decide which spike sorting tool to use and we chose wave_clus3. At the time of our experiments, the aforementioned tool convincingly demonstrated an advanced clustering performance in comparison to other algorithms including Spyking Circus, Mountain Sort, Kilosort etc.. Please also note that wave_clus3 is open source and makes no assumption on cluster shapes.

2. Example spike rasters of population activity would be very useful to visualize the temporal structure of activity across individual neurons, and also the extent to which this is correlated across time. Additionally, a summary of the distribution of units in each channel, and the change in this distribution over time would be critical.

We agree with the Reviewer and we performed new data analyses based on extracted unit data in each electrode (please check Figures 2 and 3). The activity of the extracted units was tracked over time and compared between groups. We also clustered the units based on their firing frequency and compared them between full-field stimulated and spontaneous activities. The analyses of electrode-based data were shifted to the supplementary figures (Figure S2). The raster

plots of activity in the population of selected neurons are presented in Figure 2 and Figure 3 (spontaneous, full-field evoked and holographic stimulation), as well as NBQX-APV, treated conditions in Figure S4 and S5. All related information was added to the text.

3. Clarification of network analysis due to issue 1. Based the spike sorting that has been described, this complicates the network connectivity analysis because changes in firing rate and connectivity are intermixed in their analysis. The raw traces in the channel from experiment e1 in figure 3 appear to have multiple units (look to be different waveforms). Thus, an increase in AP could be because a channel records more neurons (number of units in a channel) or a single neuron is firing more, differences it is unclear if their analysis can clarify. Additionally, single neurons can often be detected on multiple channels in MEA cultures (Chichlinsky lab, and others), and as such, functional synchrony could be a detection artifact. These issues are not clarified.

To this end, we have performed spike sorting for all recorded data. Our data show that optogenetic stimulation increased the number of active neurons (which can be reflected as increased activity in electrodes). However, probing the AP frequency in individual neurons showed a significantly elevated firing rate by optogenetic stimulation. The number of neurons firing in higher frequencies was also high in the optically stimulated neurons (Figure 2F). In standard MEAs, with a relatively large distance between two electrodes (200 μm) compared to the size of the soma (10 μm), two electrodes cannot record from the same soma. It is also very unlikely, although not impossible, that weak axonal signals within the random networks are captured by two separate electrodes of standard MEA (Dworak and Wheeler 2009 Lab Chip). Using high-density CMOS-based MEAs, one neuron can be easily detected by many electrodes. Primary neuronal cultures on high-density CMOS chips have also shown that axonal signals can be detected or tracked (Bakkum et al 2013 Nat Com). We have updated the text accordingly.

4. The burst analysis is somewhat confusing in part because of the issues about. How much of this effect can be attributed to recording from additional neurons as they mature and their responses are detected by the electrode. These details should be clarified.

After spike sorting of the recorded data, burst analyses were performed both in data from electrodes and in data from sorted neurons (Figure S2 and Figure 2, respectively). Individual neurons also showed an increase in burst frequency by optogenetic stimulation (Figure 2F). We clarified these aspects in the revised manuscript section 2.2.

5. The connectivity graphs are compelling (Fig. 6 B and C) as they appear to recapitulate work by Cossart et al on the relationship between spontaneous patterns and stimulated activity. The major claim of the authors is that the connectivity changes with development age, as illustrated in Fig. 6B. Could this be due just to spurious corrections from changes in firing rate (see Doiron et al Nature). Shuffling the spike times and then calculating the connectedness would go a long way in addressing this concern.

For Figure 6B, functional connectivity maps are calculated from the spectral entropy as described in the methods (section 3.7.9). Since this method uses filtered original data and is not based on events, a direct shuffling of the data was not possible. Furthermore, this leads to a continuous distribution of correlation coefficients between spectral entropy of different electrode signals

from which the most significant have to be chosen using a manual threshold. We tested different correlation coefficients as thresholds and selected 0.1 as the one dividing the few high correlation coefficients from a large number of low correlation coefficients. Since a shuffling of spike events is not directly feasible using this method, we decided to apply shuffling to our data by first calculating the spectral entropy over small overlapping time intervals as described in the methods and shuffling the resulting spectral entropy values of one electrode before cross-correlation with the spectral entropy of another electrode. Doing this, correlation coefficients dropped to a maximum of about 0.05 which we regard as irrelevant for further evaluation and did therefore not display here (see supplementary section 12).

Reviewer #3 (Comments to the Authors (Required)):

Combining optogenetics with neural networks in iPSC-derived organoids is a promising approach to tune and interrogate such model circuits. One common caveat of such networks is immaturity and electrogenic activity that limit their resemblance of mature physiological networks. Therefore, efforts to generate more physiological network behavior are of great value. While recently there has been some progress, this tour de force multidisciplinary study advances the field by moving toward more selective and versatile holographic optogenetic stimulation within the organoids. While holographic optogenetic stimulation has been applied in slices and intact brain, I think the scope of this study is novel and offers an avenue for how to arrive at more physiological neural networks in organoids. Given the potential of organoids for disease modeling and drug screening this advance has major relevance.

The study is quite impressive and comprehensive, the methods sound and the data of high quality. The MS has been carefully prepared. I mostly have minor edits which might help to further improve this MS.

We would like to thank the Reviewer for their positive feedback and acknowledgment of our study. We have addressed all feedback in the revised manuscript.

I think the holographic stimulation work of Valentina Emiliani and Shy Shoham deserve better representation in intro/discussion.

We would like to thank the Reviewer for pointing out the useful citations. Related works by Emiliani and Shoham have been addressed both in the introduction (paragraph 3) and as well in the discussion (Paragraph 5).

Ronzitti E, Ventalon C, Canepari M, Forget B C, Papagiakoumou E and Emiliani V (2017) Recent advances in patterned photostimulation for optogenetics *J. Opt.* **19** 113001
Shemesh O A, Tanese D, Zampini V, Linghu C, Piatkevich K, Ronzitti E, Papagiakoumou E, Boyden E S and Emiliani V (2017) Temporally precise single-cell-resolution optogenetics *Nat. Neurosci.* **20** 1796–806

Gill J V, Lerman G M, Zhao H, Stetler B J, Rinberg D and Shoham S (2020) Precise Holographic Manipulation of Olfactory Circuits Reveals Coding Features Determining Perceptual Detection *Neuron* **108** 382-393.e5

Paluch-Siegler S, Mayblum T, Dana H, Brosh I, Gefen I and Shoham S (2015) All-optical bidirectional neural interfacing using hybrid multiphoton holographic optogenetic stimulation **2** 31208

Also I found the statement starting from line 104 unclear. Moreover, the identification of the individual neurons in the present study in comparison with 2plsm imaging studies could be introduced already in the abstract and/or last section of introduction.

We added the requested information into the introduction “Holographic optogenetic stimulation” and discussion 3rd paragraph “Targeting individual neurons”.

"We added the excitatory glutamatergic synaptic blockers NBQX" consider rephrasing, e.g. glutamate receptor blocker...

We agree, “glutamatergic synaptic blockers” is now changed to “glutamate receptor blocker”.

Line 191: "poly-synaptic functional neuronal networks" there is evidence for synaptic transmission but the authors have no control on the question of across how many synapses the network is built: "synaptically connected functional neuronal networks" would probably be more appropriate

We thank the Reviewer for this comment. We applied the requested changes in the text (poly-synaptic was changed to synaptically connected networks or circuits).

"Combined holographic stimulation with MEA recordings 212 provided high spatial stimulation resolution of 8 μm spots [52]." Would be great to show some evidence for the spatial resolution here or in supp.

We agree with the Reviewer. We added Figure S10 showing a sample of the foci used for characterizing the system including Gaussian fits to determine the achieved spatial resolution.

Fig. 3: consider using blue color instead of amber for the full field, to not confuse the reader.

All colors for full-field stimulation in the figures have been changed to blue.

227: "moderate- and high-frequency APs." This should be defined

Regarding the changes in Figure 3 this part was replaced by results of neuron-based data as explained in section 2.3. Episodes that showed significant neuronal response were termed responded episodes. The description of the data was rephrased to avoid ambiguous wording.

231: "but precise AP frequency trends were revealed in our data" this is pretty useless without specifying what you are talking about

This part of Figure 3 has been based on electrode data that has been deleted and the changes have been applied in the text based on neuronal data in different episodes as well as in the raster plot. Please check Figure 3 and the corresponding text in section 2.3.

260: "A shorter delay was also attributed to a direct neuronal activation, as shown by a correlation analysis of high PSTH peak amplitudes with short latencies (Figure 4E)." this notion would be best asserted by recordings with NBQX+APV, the statement of line 264 should only be made if documented by this experiment.

The results of NBQX treated conditions have been analyzed both at the electrode level (Figure S4) and in the sorted neurons (Figure S5). We also extracted PSTH plots both in electrode data and as well in detected neurons. Based on PSTH in (Figure S4D), evoked activity in electrodes last 200 ms (150 ms beyond applied 50ms pulses). The first 50 ms of electrode activity showed a high PSTH peak and timely reproducible responses to all applied pulses. However, delayed electrode activity after applied pulses expressed lower PSTH amplitude. The delayed responses to different pulses were not timely reproducible. These delayed responses disappeared in the presence of NBQX-APV, which means they are mainly derived from synaptic communication. We observed a similar trend in the neurons that reproducible part of the responses remained intact but the delayed responses have been erased in presence of NBQX-APV. Unfortunately, it was not possible to run this experiment in the holographic stimulation setup because in the presence of NBQX-APV the whole network was mainly silent and the number of neurons responding to light stimuli was very low. Even with higher intensities of full-field stimulation we observed limited neuronal responses in the presence of NBQX-APV. Delayed responses to the full-field stimulation showed a similar profile as delayed responses to the holographic stimulation: lower PSTH amplitude with non-reproducible responses across applied pulses (that is visible in the raster plot of responses in the delayed phase). Altogether, these data show that delayed neuronal responses (delayed PSTH peak) are mainly derived from synaptic communication rather than its direct response to the light pulse.

Section 2.5 needs some work: if I get it right the authors perform in-depth analysis of a single recording with exemplary responses to then arrive at the conclusion that holographic optogenetic stimulation but not full-field optogenetic stimulation revealed a specific motif.

An exemplar motif analysis was performed based on holographic stimulation data. This was not feasible with full-field stimulation that simultaneously activates all neurons. We explained this by statistical means in Figure 6 in which we report the overall network connectivity maps derived from holographic stimulation data and as well the connection number and strength. The overall map has been assembled by pooling all connectivity motifs together as one graph. To prevent confusion the description of details regarding an exemplar motif in Figure 5 has been included in the figure legend. Explanation of the corresponding results was modified in the text at section 2.5.

While I understand that the authors are excited here, I do find this a bit weak as evidence supporting their conclusion. The authors need to provide additional data to support the notion of direct and indirect stimulation and might want to use NBQX+APV exposure as further test

The extracted PSTH and raster plots based on electrode data or neuronal unit activity showed that the delayed tail of the evoked responses (150 ms after applied light pulse) disappeared by NBQX-APV treatment and blocking the glutamatergic receptors (Figure S4D and Figure S5D). This indicates that directly stimulated neurons evoke an indirect response in other neurons through excitatory synapses. As explained before it was not possible to run this experiment in holographic stimulation setup. However, the PSTH data obtained from full-field stimulation in presence of NBQX-APV was compatible with holographic data regarding the highly coupled early responses during applied pulses vs. non-reproducible late responses that appeared after applied pulses.

March 23, 2022

RE: Life Science Alliance Manuscript #LSA-2021-01268R

Prof. Volker Buskamp
Universitäts-Augenklinik Bonn
Dep. of Ophthalmology
Ernst-Abbe-Straße 2
Bonn 53127
Germany

Dear Dr. Buskamp,

Thank you for submitting your revised manuscript entitled "Tracking connectivity maps in human stem-cell-derived neuronal networks by holographic-optogenetics". We would be happy to publish your paper in Life Science Alliance pending final revisions necessary to meet our formatting guidelines.

- please add your main, supplementary figure, and table legends to the main manuscript text after the references section;
- we encourage you to revise the figure legends for figures S5 such that the figure panels are introduced in alphabetical order;
- Please upload all figure files as individual ones, including the supplementary figure files; all figure legends should only appear in the main manuscript file
- please add ORCID ID for secondary corresponding author-he should have received instructions on how to do so
- please add a conflict of interest statement to your main manuscript text
- please use the [10 author names, et al.] format in your references (i.e. limit the author names to the first 10)
- please add callouts for Figures S2A-B; S3A-B; S7A-D to your main manuscript text
- please add Data Availability section

A. FINAL FILES:

B. MANUSCRIPT ORGANIZATION AND FORMATTING:

Sincerely,

Reviewer #1 (Comments to the Authors (Required)):

The authors have addressed most of our suggestions and comments. Overall, I believe that the study will be valuable to the community.

Reviewer #2 (Comments to the Authors (Required)):

The authors have addressed my concerns.

Reviewer #3 (Comments to the Authors (Required)):

The authors have addressed most of my comments. I support publication.

March 29, 2022

RE: Life Science Alliance Manuscript #LSA-2021-01268RR

Prof. Volker Buskamp
Universitäts-Augenklinik Bonn
Dep. of Ophthalmology
Ernst-Abbe-Straße 2
Bonn 53127
Germany

Dear Dr. Buskamp,

Thank you for submitting your Methods entitled "Tracking connectivity maps in human stem-cell-derived neuronal networks by holographic-optogenetics". It is a pleasure to let you know that your manuscript is now accepted for publication in Life Science Alliance. Congratulations on this interesting work.

DISTRIBUTION OF MATERIALS:

Again, congratulations on a very nice paper. I hope you found the review process to be constructive and are pleased with how the manuscript was handled editorially. We look forward to future exciting submissions from your lab.

Sincerely,
